# F$^3$Set: Towards Analyzing Fast, Frequent, and Fine-grained Events from Videos

**Zhaoyu Liu[1,2], Kan Jiang[2], Murong Ma[2], Zhe Hou[3], Yun Lin[4], Jin Song Dong[2]**

[1]Ningbo University [2]National University of Singapore [3]Griffith University
[4]Shanghai Jiao Tong University
{liuzy, jiangkan}@nus.edu.sg, murongma@u.nus.edu
z.hou@griffith.edu.au, lin_yun@sjtu.edu.cn, dcsdjs@nus.edu.sg

## Abstract

Analyzing Fast, Frequent, and Fine-grained (F$^3$) events presents a significant challenge in video analytics and multi-modal LLMs. Current methods struggle to identify events that satisfy all the F$^3$ criteria with high accuracy due to challenges such as motion blur and subtle visual discrepancies. To advance research in video understanding, we introduce F$^3$Set, a benchmark that consists of video datasets for precise F$^3$ event detection. Datasets in F$^3$Set are characterized by their extensive scale and comprehensive detail, usually encompassing over 1,000 event types with precise timestamps and supporting multi-level granularity. Currently F$^3$Set contains several sports datasets, and this framework may be extended to other applications as well. We evaluated popular temporal action understanding methods on F$^3$Set, revealing substantial challenges for existing techniques. Additionally, we propose a new method, F$^3$ED, for F$^3$ event detections, achieving superior performance. The dataset, model, and benchmark code are available at https://github.com/F3Set/F3Set.

## 1 Introduction

Recognizing sequences of fast (fast-paced), frequent (many actions in a short period), and fine-grained (diverse types) events with precise timestamps (with a tolerance of 1-2 frames) is a challenging problem for both current video analytics methods and multi-modal large language models (LLMs). Despite advances in fine-grained action recognition [31; 58; 51], temporal action localization [60; 6; 40; 59], segmentation [67; 33; 71; 2], and video captioning [63; 56; 49; 36], limited focus has been focused on this problem. This task is critical for various real-world applications, such as sports analytics, where action forecasting [21; 65], strategic and tactical analysis [44; 45; 46; 48], and player performance evaluation [10; 55] depend on understanding *detailed* of event sequences. Other examples include industrial inspection [42], crucial for detecting subtle irregularities in high-speed production lines to ensure quality and safety; computer vision in autonomous driving [25], essential for accurate and instantaneous vehicle control and obstacle detection; and surveillance [53], important for the precise identification of abnormal or sudden events to enhance security. However, existing methods and datasets foundational to their development only *partially* address the F$^3$ scenario.

To facilitate the study of F$^3$ events understanding, we propose a new benchmark, F$^3$Set, for precise temporal events detection and recognition. F$^3$Set datasets usually have a large number of event types (on the order of 1,000), annotated with exact timestamps, and offer multi-level granularity to capture comprehensive event details. Although F$^3$ is a general problem, creating such a dataset requires domain-specific knowledge for labeling and processing; thus, in this paper, we use tennis as a case study. We also introduce a general annotation pipeline and toolchain to support domain experts in creating new F$^3$ datasets. Using this pipeline, we have also been building datasets for table tennis and badminton, and a community of users is actively expanding these with other applications.

Unlike other video analysis tasks, tennis actions are characterized by their rapid succession and diversity, as illustrated in Figure 1. Understanding detailed event attributes such as shot direction, technique, and outcome is essential. For example, analyzing patterns in serve directions (e.g., "T", "body", "wide", defined in Appendix B) or success rates can reveal players' habits and skill levels, offering strategic insights for competitive advantage [15]. This detailed analysis supports coaches and players in developing tailored strategies against different opponents [16; 47]. However, detecting

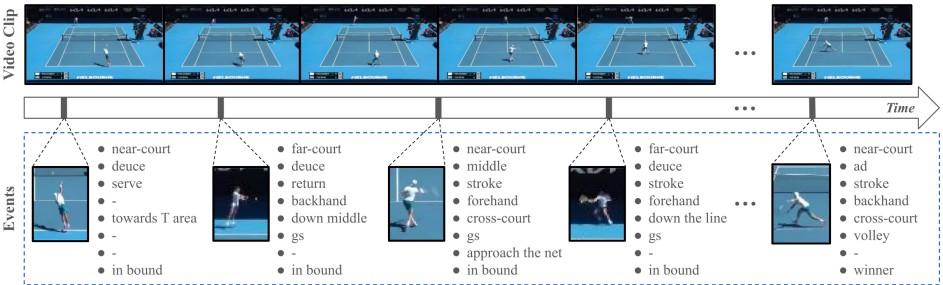

Figure 1: Example of detecting fast, frequent, and fine-grained events with precise moments.

$F^3$ events from videos poses significant challenges, such as subtle visual differences, motion-induced blurring, and the need for precise event localization. Current video understanding methods are inadequately equipped to address these challenges. For instance, traditional fine-grained action recognition [9; 58; 28] assigns a single label to an entire video rather than identifying a sequence of events. Temporal action localization (TAL) and temporal action segmentation (TAS) often depend on pre-trained or modestly fine-tuned input features [39; 14], which lack the specificity required to capture the subtle and domain-specific visual details necessary for recognizing diverse events with temporal precision. Some studies [24; 36; 43] attempt to address these issues through *dense* frame sampling and end-to-end training. However, this makes targeted events temporally sparse (e.g., only a few events over hundreds of consecutive frames). As a result, long-term temporal correlation modules on dense visual features struggle to capture event-wise causal correlations effectively.

Moreover, Large Language Models (LLMs) [54; 61; 38] have expanded their capabilities to include multi-modal inference, encompassing text, visuals, and audio. Recognizing the potential, we conducted preliminary experiments on $F^3$Set using GPT-4 and observed that it understood basic video contexts, such as sports types, contextual information (e.g., court type and scoreboard), and simple actions. However, it struggles with understanding $F^3$ events and temporal relations between frames (e.g., shot directions). See Appendix A for details. Consequently, GPT-4 yields poor results compared to the other methods for $F^3$ problems, and we do not use it in the experiment. By introducing $F^3$Set, we hope it can help advance multi-modal LLM capabilities in $F^3$ video understanding in the future.

Leveraging $F^3$Set, we extensively evaluate existing temporal action understanding methods, aiming to reveal the challenges of $F^3$ event understanding. To provide guidelines for future research, we conduct a number of ablation studies on modeling choices. Addressing the shortcomings of existing methods, we also propose a simple yet efficient model, $F^3$ED, that is designed for $F^3$ event detection tasks and can be trained quickly on a single GPU. It outperforms existing models and can serve as a baseline for further development.

**Contributions.** The key contributions of this paper are as follows:

- We create $F^3$Set, a new benchmark with datasets that feature over 1,000 precisely timestamped event types with multi-level granularity, designed to challenge and advance the state-of-the-art in temporal action understanding.
- We introduce a general annotation toolchain that enables domain experts to create new $F^3$ datasets.
- We propose an end-to-end model named $F^3$ED, which can accurately detect $F^3$ event sequences from videos through visual features and contextual sequence refinement on a single GPU.
- We assess the performance of leading temporal action understanding methods on $F^3$Set through comprehensive evaluations and ablation studies and analyze the results.

## 2 RELATED WORK

**Existing $F^3$ related datasets.** Although datasets have been developed for temporal action understanding, few focus on the $F^3$ events. Table 1 compares existing datasets with $F^3$Set by scale ("# Vid", "# Clips") and characteristics like action speed ("Evt. Len."), frequency ("Evt. / sec"), and granularity ("# Classes"), which correspond to "fast", "frequent", and "fine-grained" respectively. Datasets such as THUMOS14 [27] and Breakfast [30] focus on coarse-grained actions, where background context provides clear cues, and actions span seconds to minutes. In contrast, FineAction [41] and ActivityNet [4] cover a wide range of daily activities with diverse action categories, while FineGym [58] delves into detailed action types within gymnastics. Like FineGym, $F^3$Set emphasizes domain-specific

Table 1: Comparison of existing $F^3$ related datasets and $F^3$Set. "Evt. Len." is the average duration of each event, and "# Evt. / sec" is the average number of events per second.

| Datasets | # Vid. | # Clips. | Avg. Clip Len. | # Classes | Evt. Len. | # Evt. / sec |
|---|---|---|---|---|---|---|
| *(a) Fine-grained* | | | | | | |
| FineAction [41] | - | 16,732 | 149.5s | 101 | 6.9s | 0.3 |
| ActivityNet [4] | - | 19,994 | 116.7s | 200 | 49.2s | 0.01 |
| FineGym [58] | 303 | 32,697 | 50.3s | 530 | 1.7s | 0.3 |
| *(b) Fast* | | | | | | |
| CCTV-Pipe [42] | 575 | 575 | 549.3s | 16 | < 0.1s | 0.02 |
| SoccerNetV2 [11] | 9 | 9 | 99.6min | 12 | < 0.1s | 0.3 |
| *(c) Frequent* | | | | | | |
| FineDiving [69] | 135 | 3,000 | 4.2s | 29 | 1.1s | ~1 |
| *(d) Fast & Frequent* | | | | | | |
| ShuttleSet [66] | 44 | 3,685 | 10.9s | 18 | < 0.1s | ~1 |
| $P^2$ANet [3] | 200 | 2,721 | 360.0s | 14 | < 0.1s | ~2 |
| *(d) Fast & Frequent & Fine-grained* | | | | | | |
| **$F^3$Set** | 114 | 11,584 | 8.4s | 1,108 | < 0.1s | ~1 |

granularity with subtle visual differences but encounters additional challenges due to faster and more frequent actions. Besides, unlike FineGym's typical single-player focus, $F^3$Set (e.g., tennis) features two players and a fast-moving ball, with both players rapidly moving across the court, occupying only small portions of the scene, thus increasing task difficulty. CCTV-Pipe [42] targets temporal defect detection in urban pipe systems, providing single-frame annotations for rapid event detection, though it is limited in frequency and event types. Research in the sports domain has explored the detection of fast and frequent actions. FineDiving [69] segments diverse diving events, while ShuttleSet [66] and $P^2$ANet [3] focus on identifying strokes in fast-paced racket sports. Volleyball [26] and NSVA (basketball) [68] focus on team sports understanding and video captioning, while SoccerNetV2 [11] ball action spotting task focus on identifying the timing and type of ball-related actions. However, these datasets typically cover coarser event types and are limited to specific $F^3$ aspects.

In contrast, our proposed $F^3$Set is characterized by 1) *rapid* events that occur instantaneously, 2) *high frequency* of approximately one event per second, and 3) *extensive granularity* with a larger number of detailed event classes. These attributes introduce novel challenges.

**$F^3$ event understanding** Detecting $F^3$ events poses unique challenges due to their rapid temporal dynamics, high occurrence rates, and subtle visual distinctions, requiring precise temporal and contextual understanding. Fine-grained action detection has been explored in tasks covering diverse daily activities [4; 41], using features extracted by video encoders pre-trained on datasets like Kinetics-400 [29] and a detection head for classification. However, such pre-trained extractors often miss domain-specific nuances. Domain-specific methods in FineGym [58] and FineDiving [58] utilize end-to-end training to incorporate domain knowledge. These methods often encode videos into non-overlapping snippets or downsample frames, yielding coarse temporal features insufficient for detecting fast-paced events spanning only 1–2 frames. Related works such as ShuttleSet [66] and $P^2$ANet [3] address fast and frequent event detection in racket sports by employing end-to-end models that extract frame-wise features and use detection heads (e.g., BMN [37] or GRU [8]) to classify each frame. To address class imbalance, the loss weight of the foreground classes is set higher than the background during training [24]. While these approaches achieve precise temporal spotting, their scalability to larger action classes is limited by challenges like long-tail class distributions and inadequate modeling of event-wise correlations. Our proposed $F^3$ED overcomes these issues through frame-wise dense processing, a multi-label classification head to handle minor event differences and class imbalances, and a contextual module to refine predictions by leveraging event-wise causal relationships, enhancing both precision and robustness in $F^3$ event detection.

## 3 $F^3$SET: A BENCHMARK DATASET FOR $F^3$ EVENT DETECTION

Recognizing the limitations in existing video datasets for $F^3$ event understanding, we introduce $F^3$Set, a new benchmark for precise temporal $F^3$ events detection and recognition. Given the need for

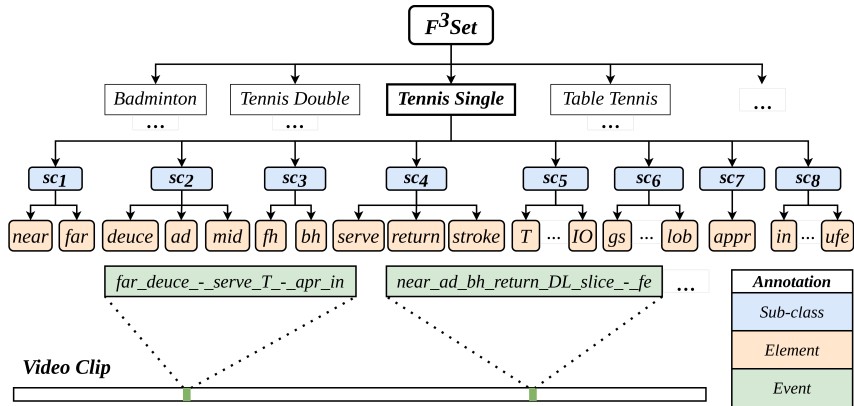

Figure 2: Breakdown of $F^3$Set event class annotation.

domain-specific expertise in creating $F^3$ datasets, this section uses **tennis** as a **case study** to illustrate $F^3$Set's event descriptions, construction process, and key properties. We also propose a general annotation pipeline and toolchain that empowers domain experts to develop new $F^3$ datasets for diverse applications. Applying the same approach, we have also built $F^3$ datasets for **other domains**, including tennis doubles, badminton, and table tennis (see link).

## 3.1 $F^3$SET EVENT DESCRIPTION

We use **tennis** to illustrate $F^3$ event descriptions, introducing key lexicon and defining $F^3$ events. Datasets have been built for **other $F^3$domains**, including tennis doubles, badminton, and table tennis, with similar event definitions. Details are in Appendix C.

**Lexicon.** A tennis court is divided into deuce, middle, and ad regions. The initial shot, a "serve," targets the T, Body (B), or Wide (W) areas. A "return" follows if the receiver's shot lands in bounds. Subsequent shots, or "strokes", can be directed "cross-court" (CC), "down the line" (DL), "down the middle" (DM), "inside-in" (II), or "inside-out" (IO) using either "forehand" (fh) or "backhand" (bh). Players may "approach" (apr) the net on shorter balls. Shot techniques include "ground stroke/top spin" (gs), "slice", "volley", and "lob", with outcomes: "in-bound", "winner", "forced error", or "unforced error". More detailed definitions can be found in Appendix B.

**$F^3$ events.** Formally, each event consists of 8 *sub-classes*, denoted as $sc_1, sc_2, ..., sc_8$:

$sc_1$ – *hit by which player*: (1) near- or (2) far-end player;

$sc_2$ – *hit from which court location*: (3) deuce, (4) middle, or (5) ad court;

$sc_3$ – *hit at which side of the body*: (6) forehand or (7) backhand;

$sc_4$ – *shot type*: (8) serve, (9) return, or (10) stroke;

$sc_5$ – *shot direction*: (11) T, (12) B, (13) W, (14) CC, (15) DL, (16) DM, (17) II, or (18) IO;

$sc_6$ – *shot technique*: (19) gs, (20) slice, (21) volley, (22) lob, (23) drop, or (24) smash;

$sc_7$ – *player movement*: (25) approach;

$sc_8$ – *shot outcome*: (26) in, (27) winner, (28) forced error, or (29) unforced error.

Altogether, there are 29 *elements* and 1,108 *event types* based on various combinations (Figure 2).

Similarly, for other domains, badminton contains 6 *sub-classes*, 28 *elements* and 1008 *event types*; table tennis contains 7 *sub-classes*, 23 *elements* and 1296 *event types*; and tennis doubles contain 26 *elements* and 744 *event types*. Compared to existing racket sports video datasets [3; 66], $F^3$Set offers additional dimensions, such as shot direction and outcomes, which are crucial for identifying playing patterns and success rates. Please refer to Appendix C for more details.

## 3.2 $F^3$SET DATASET CONSTRUCTION

**Video collection.** For tennis, we collected publicly available high-resolution singles matches (2012–2023) from YouTube, including Grand Slams, Olympics, and major ATP/WTA tournaments. The dataset includes various court surfaces (hard, clay, grass), male and female players, and both right- and left-handed competitors. These videos feature complete rallies, match footage, and detailed player data. Similar criteria were used for tennis doubles, badminton, and table tennis videos.

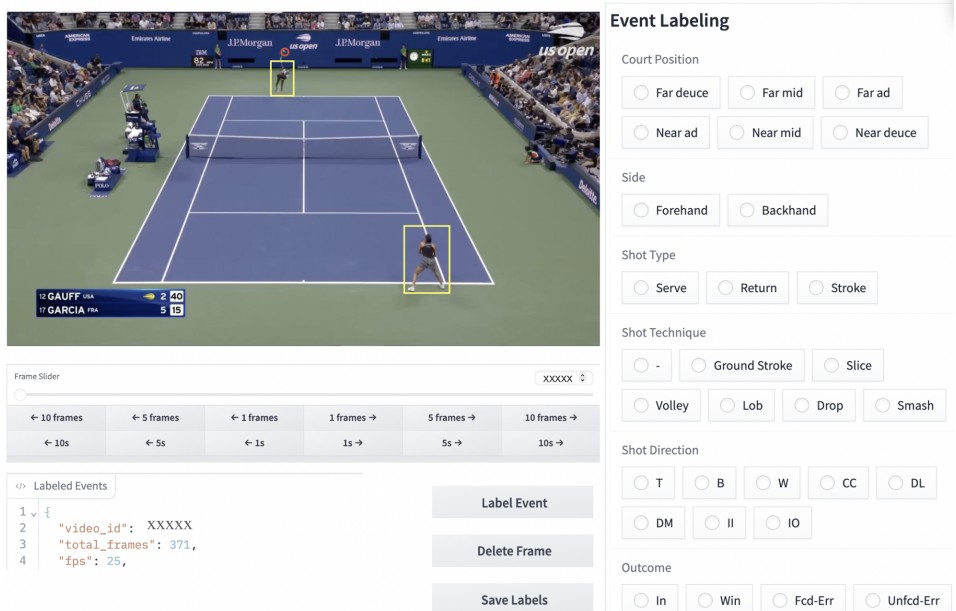

Figure 3: An interface of the labeling tool. The panel on the right is application-customizable.

**Annotation pipeline and toolchain.**    After data collection, we use a three-stage annotation process designed to maximize automation and minimize manual effort. This pipeline is adaptable to various sports broadcast videos and broader domains:

*(1) Video segmentation*: The first stage is to segment a full broadcast video into shorter clips using a context-aware scene detector [1] that automatically identifies jump cuts within the video.

*(2) Clip selection*: The second stage is to select targeted clips (e.g., clips contain tennis rallies) using a Siamese network to compare each clip with a "base image" indicative of the scene of interest.

*(3) $F^3$ event annotation*: The final stage is to identify the precise event moments (e.g., frames when a player hits the ball) and record the corresponding event types through an annotation tool.

The first two steps are automated and applicable to a range of sports videos, facilitating the efficient breakdown of lengthy videos into relevant clips. For the final phase, we developed an interactive annotation interface, shown in Figure 3. The tool allows users to navigate clips quickly (e.g., 1-second increments) or review them frame by frame, enabling efficient identification of key events (e.g., hitting moments). It supports selecting shot types and identifying court positions through direct clicks on the video, with each click displayed for immediate verification. Object-level detection can assist the process, and a foolproof design minimizes errors from accidental clicks or misjudgments. This tool is adaptable to other sports by incorporating domain-specific knowledge, broadening its applicability.

Our annotation team consists of 8 members. We provided them with specialized training and rigorous pre-tests before beginning the official annotation work, along with supporting materials such as slides and demonstrations. Each annotator was assigned an equal portion of the dataset, totaling 1,450 clips (rallies) each. The manual labeling takes roughly 30 hours to finish all 1,450 clips. Following the initial annotation phase, we conducted multiple rounds of cross-validation involving random sampling of rallies and quality checks among annotators to ensure the accuracy of the event-based labels. In cases where conflicting annotations arose, annotators were asked to input the labels they believed to be correct. The final label was determined based on a majority vote among the annotators.

### 3.3   $F^3$SET DATASET STATISTICS AND PROPERTIES

Key statistics for $F^3$Set tennis dataset are summarized in Table 2. Statistics for other $F^3$ datasets, including badminton, table tennis, and tennis doubles, are provided in the Appendix D. We employ a training, validation, and testing split of 3:1:1, with the training and validation sets drawn from the same video sources, while the test set features clips from distinct videos.

**Event Timestamp.**    Unlike typical TAL and TAS tasks, where an action spans several frames or seconds, the duration of actions in racket sports is often ambiguous. Thus, stroke actions are defined as instantaneous events, recording only the moment of ball-racket contact [62] as shown in Figure 1.

Table 2: Summary of F$^3$Set tennis dataset statistics.

| Category | Details | | Category | Details |
|---|---|---|---|---|
| Matches | 114 broadcast matches | | Clips | 11,584 rallies |
| Players | 75 (30 men, 45 women) | | Avgerage Clip Duration | 8.4 sec |
| Handedness | 68 right-handed, 7 left-handed | | Total Shots | 42,846 |
| Frame Rate (FPS) | 25–30 FPS | | Shots Per Rally | 1 to 34 |

**Multi-level granularity.** Depending on the requirements of the analytics task, F$^3$Set can focus on a subset of sub-classes, enabling flexible granularity. We define a parameter $G \in \mathcal{P}(\{sc_1, \ldots, sc_8\})$, where $\mathcal{P}(\{sc_1, \ldots, sc_8\})$ is the power set of $\{sc_1, \ldots, sc_8\}$, to select sub-classes and form different levels of granularity. We define 3 granularity levels using F$^3$Set tennis as an example. At the coarse level, $G_{\text{low}} = \{sc_1, sc_3, sc_4, sc_8\}$ includes 4 sub-classes, 11 elements, and 38 event types. This level captures essential but broad information. At a finer level, $G_{\text{mid}} = \{sc_1, \ldots, sc_6\}$ consists of 6 sub-classes, 24 elements, and 365 event types. This granularity provides more detailed event representations. At the most detailed level, $G_{\text{high}} = \{sc_1, \ldots, sc_8\}$ encompasses all 8 sub-classes, 29 elements, and 1,108 event types. This level is ideal for precise and comprehensive event analysis. This multi-level granularity enhances F$^3$Set's flexibility for diverse real-world tasks.

### 3.4 ETHICAL CONSIDERATIONS

F$^3$Set is constructed from publicly available sports broadcasts, ensuring compliance with ethical and legal standards. We do not redistribute video content, providing only YouTube links to maintain adherence to copyright policies. The dataset focuses on professional players in public tournaments, avoiding private or off-court data and ensuring it is used strictly for academic research. While anonymization is not applied, as these players are public figures, we emphasize that the dataset should not be used for non-research purposes. A more detailed discussion on privacy, consent, and bias mitigation is provided in Appendix E.

## 4 OUR PROPOSED APPROACH: F$^3$ED

Acknowledging the challenges and limitations of existing approaches, we propose a simple yet effective method named **F**ast **F**requent **F**ine-grained **E**vent **D**etection network (F$^3$ED), illustrated in Figure 4. It is designed for F$^3$ event detection and can serve as a baseline for further development.

**Problem formulation.** Let $X \in \mathbb{R}^{H \times W \times 3 \times N}$ denote the input, consisting of $N$ RGB frames of size $H \times W$. The output is a sequence of $M$ event-timestamp pairs $((E_1, t_1), \ldots, (E_M, t_M))$, where $E_i$ is the event type with $C$ classes and $t_i$ is the corresponding timestamp for $i \in \{1, \ldots, M\}$. Additionally, each event $E_i$ can also be expressed as a vector $[e_{i,1}, \ldots, e_{i,K}]$, with each element $e_{i,j} \in \{0, 1\}$ indicating the presence or absence of the $j^{th}$ element in event $E_i$, where $j$ is an integer $j \in \{1, \ldots, K\}$. The parameter $K$, which defines the *number of elements* in each event vector.

**Video Encoder (VE).** The first stage of both baselines and our model will extract spatial-temporal frame-wise features. The video encoder (VE) consists of a visual backbone, followed by a bidirectional GRU to capture long-term visual dependencies: $\mathbf{F}_{emb} = \text{VE}(X)$, with $\mathbf{F}_{emb} \in R^{N \times d'}$.

**Event Localizer (LCL).** Utilizing the frame-wise features $\mathbf{F}_{emb}$, the event localizer (LCL) employs a fully connected network with a Sigmoid activation function to perform dense binary classification, aiming to accurately identify specific event instances. For an $N$-frame clip, the output is represented as $(\hat{p}_1, \ldots, \hat{p}_N)$, where each $\hat{p}_i$ denotes the probability that an event occurs at the corresponding timestamp: $(\hat{p}_1, \ldots, \hat{p}_N) = Sigmoid(\text{LCL}(\mathbf{F}_{emb}))$. Ground truth labels $(p_1, \ldots, p_N)$ with $p_i \in \{0, 1\}$ are used to compute the discrepancy between the predicted probabilities and the actual values using binary cross-entropy loss as: $L_{LCL} = \frac{1}{N} \sum_{i=1}^{N} p_i \cdot log(\hat{p}_i) + (1 - p_i) \cdot log(1 - \hat{p}_i)$.

**Multi-label Event Classifier (MLC).** Upon detecting events, we proceed to categorize them into specific types using a multi-label classification module (MLC). This module, a fully connected network, takes the identified event features $f_i$ from $\mathbf{F}_{emb}$ as inputs to predict the event types: $\hat{E}_i = Sigmoid(\text{MLC}(f_i)) = [\hat{e}_{i,1}, \ldots, \hat{e}_{i,K}]$, where $K$ denotes the number of elements, $f_i$ represents the features for the event at the $i^{th}$ frame, $\hat{E}_i$ is the predicted event type, and $\hat{e}_{i,j} \in [0, 1]$ is the probability of $\hat{E}_i$ containing the $j^{th}$ element. For a video clip with $M$ events, the ground truths are

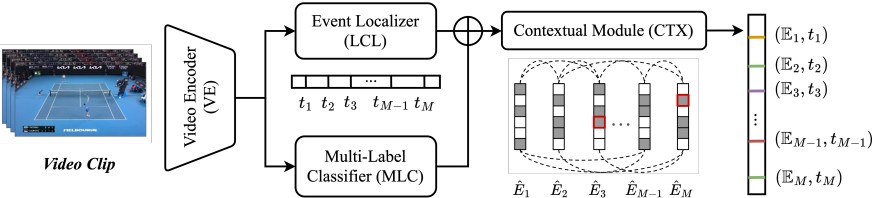

Figure 4: Overview of F³ED. RGB images are processed by VE to capture frame-wise spatial-temporal features, which are passed to LCL to identify event timestamps and MLC to predict labels. Outputs from LCL and MLC are combined ('plus' symbol) to form an event representation sequence and refined by CTX module. 'Red squares' represent errors from purely visual predictions.

given as $(E_1, \ldots, E_M)$ with each $E_i$ represented as a vector of $K$ elements $[e_{i,1}, \ldots, e_{i,K}]$. The loss can be represented by $L_{MLC} = \frac{1}{M} \sum_{i=1}^{M} (\frac{1}{K} \sum_{j=1}^{K} e_{i,j} \cdot log(\hat{e}_{i,j}) + (1 - e_{i,j}) \cdot log(1 - \hat{e}_{i,j}))$.

**Contextual module (CTX)**    Video encoders often struggle to extract insightful visual features from fast-paced videos due to motion blur, and objects of interest, such as players, may only occupy a small portion of the frame. This can result in the loss of crucial visual details for fine-grained action classification, particularly when resizing images to $224 \times 224$. Selecting the best-predicted event types naively might, therefore, produce invalid event sequences. To address this, we introduce a contextual module (CTX), designed to concurrently learn contextual knowledge from event sequences during end-to-end training: $(\mathbb{E}_1, \ldots, \mathbb{E}_M) = \text{CTX}(\hat{E}_1, \ldots, \hat{E}_M)$. CTX employs a bidirectional GRU to process the predicted event sequence $\hat{E}$ and outputs a refined sequence $\mathbb{E}_i = [\mathbb{e}_1, \ldots, \mathbb{e}_k]$, integrating both visual-based predictions and contextual correlations across events. The loss is calculated for each refined event: $L_{CTX} = \frac{1}{M} \sum_{i=1}^{M} (\frac{1}{K} \sum_{j=1}^{K} e_{i,j} \cdot log(\mathbb{e}_{i,j}) + (1 - e_{i,j}) \cdot log(1 - \mathbb{e}_{i,j}))$.

## 5 EXPERIMENTS

In this section, we benchmark existing temporal action understanding methods, including TAL, TAS, and TASpot, on the F³Set dataset and conduct a series of ablation studies.

**Evaluation metrics.**    The evaluation metrics used in our work are carefully chosen to comprehensively assess both the temporal precision and classification accuracy of detected events, which are critical for F³ event detection. These metrics align with evaluation standards in similar tasks. **Edit Score** [32] measures the similarity between predicted and ground truth event sequences using Levenshtein distance, capturing errors in event sequence structure, such as missing, additional, or misordered events. This metric is particularly valuable for evaluating models where the temporal order and completeness of event sequences are essential [23]. **Mean F1 Score with Temporal Tolerance** evaluates both classification and temporal localization accuracy [24; 23]. By considering a prediction correct only when its timestamp aligns within a strict temporal tolerance (e.g., $\pm 1$ frame) and its class correctly identifies, this metric ensures that models are assessed on their ability to achieve precise temporal spotting alongside accurate classification. Given the long-tail distribution of event types in the dataset, where some events are extremely rare, we report two variants of the mean F1 score to ensure a balanced evaluation: $F1_{evt}$, the average F1 score across all event types, and $F1_{elm}$, the average F1 score across all elements, which typically presents a more balanced distribution.

**Baselines.**    Existing temporal action understanding frameworks typically incorporate two key components: a *video encoder* for visual feature extraction and a *head module* for specific tasks such as detection or segmentation. Applying these models directly to our study presents challenges, as they generally utilize a two-stage training process—employing a static, pre-trained video encoder for feature extraction and training only the head module. This approach often fails to capture fine-grained, domain-specific events due to its reliance on temporally coarse, non-overlapping, or downsampled video segments. To address these limitations, we have adapted these temporal action understanding methods to develop new baselines better suited for detecting F³ events. Given the rapid pace and short duration of tennis shots, it is crucial to utilize frame-wise feature extraction [7] (discussed in Section 5.2). Besides, end-to-end training with video encoder fine-tuning is required to capture the subtle event differences. Moreover, the classification of some sub-classes (e.g., shot direction, outcome) demands long-term temporal reasoning to integrate information from subsequent frames.

Consequently, we focus on established feature extractors: TSN [64], SlowFast [20], I3D [5], VTN [52], and TSM [35], which enable frame-wise feature extraction and end-to-end training. We pair

Table 3: Experimental results on $F^3$Set (tennis) with 3 levels of granularity. Full table in Appendix G.

| Video encoder | Head arch. | $F^3$Set ($G_{high}$) | | | $F^3$Set ($G_{mid}$) | | | $F^3$Set ($G_{low}$) | | |
|---|---|---|---|---|---|---|---|---|---|---|
| | | $F1_{evt}$ | $F1_{elm}$ | Edit | $F1_{evt}$ | $F1_{elm}$ | Edit | $F1_{evt}$ | $F1_{elm}$ | Edit |
| TSN [64] | MS-TCN [19] | 15.9 | 59.8 | 53.5 | 23.2 | 60.9 | 65.8 | 45.7 | 70.4 | 72.8 |
| | ActionFormer [72] | 18.4 | 60.6 | 55.2 | 24.8 | 61.9 | 67.3 | 48.7 | 70.6 | 72.2 |
| | E2E-Spot [24] | 24.7 | 65.3 | 60.1 | 31.5 | 66.2 | 71.0 | 53.5 | 73.6 | 75.0 |
| SlowFast [20] | G-TAD [70] | 23.0 | 66.1 | 64.0 | 29.6 | 66.5 | 74.2 | 53.3 | 76.0 | 77.9 |
| | ActionFormer [72] | 28.7 | 70.0 | 67.6 | 35.5 | 70.9 | 76.4 | 59.3 | 77.1 | 81.5 |
| | E2E-Spot [24] | 25.9 | 69.4 | 65.7 | 33.8 | 70.4 | 75.4 | 55.5 | 76.5 | 79.5 |
| I3D [5] | E2E-Spot [24] | 22.7 | 59.7 | 68.7 | 27.1 | 60.7 | 74.2 | 51.9 | 67.7 | 78.3 |
| VTN [52] | E2E-Spot [24] | 14.8 | 58.3 | 56.7 | 20.0 | 59.4 | 68.2 | 39.7 | 63.1 | 73.1 |
| TSM [35] | MS-TCN [19] | 21.7 | 67.3 | 58.6 | 30.4 | 69.5 | 73.0 | 50.2 | 74.0 | 75.3 |
| | ASformer [71] | 17.6 | 61.9 | 57.5 | 25.5 | 64.0 | 74.2 | 46.0 | 72.9 | 74.0 |
| | G-TAD [70] | 16.9 | 62.5 | 55.2 | 29.8 | 66.9 | 74.8 | 39.8 | 70.1 | 67.2 |
| | ActionFormer [72] | 22.4 | 65.7 | 60.3 | 31.0 | 68.2 | 74.7 | 52.4 | 73.8 | 74.9 |
| | E2E-Spot [24] | 31.4 | 71.4 | 68.7 | 39.5 | 72.3 | 77.9 | 60.6 | 78.4 | 82.1 |
| TSM[35] | $F^3$ED | **40.3** | **75.2** | **74.0** | **48.0** | **76.5** | **82.4** | **68.4** | **80.0** | **87.2** |

each encoder with five representative head module architectures from existing methods: MS-TCN [19] and ASFormer [71] from TAS, G-TAD [70] and ActionFormer [72] from TAL and E2E-Spot [24] from TASpot, to establish a set of new baseline models for our study. To identify hitting moments and their respective event types, frame-wise dense *multi-class* classification is applied to identify each frame as either background or one of the event types.

**Implementation details.**   We implement and train models on $F^3$Set in an end-to-end manner. The video encoder takes video clip $X$ down-scaled and cropped to $224 \times 224$ to extract frame-wise visual features. Subsequently, each head module processes per-frame features to identify a sequence of $F^3$ events and their timestamps. For more implementation details, please refer to Appendix F.

### 5.1 RESULTS AND ANALYSIS

The evaluation results presented in Table 3 provide several critical insights into the performance of various methods across different levels of granularity ($G_{low}$, $G_{mid}$, and $G_{high}$). A general trend emerges where performance decreases as granularity increases, underscoring the growing challenges associated with finer granularity. While certain methods demonstrate some robustness, the overall efficacy across all approaches remains suboptimal, particularly at higher levels of granularity, indicating the challenge of precise $F^3$ event detection task.

Simple 2D CNNs (e.g., TSN), which process frames independently, are inadequate for $F^3$ event detection due to their inability to capture critical spatial-temporal correlations between frames. Lacking temporal modeling, they struggle to distinguish visually similar events, resulting in poor performance, especially at higher granularity levels. Advanced video encoders such as I3D [5], SlowFast [20], and transformer-based VTN [52], which excel in other video understanding tasks, face significant challenges with $F^3$Set. These models process video data using techniques like non-overlapping snippets or frame downsampling, resulting in coarse temporal features. While effective for long-duration actions, such approaches struggle to detect the rapid, short-duration events in $F^3$, which rely on precise temporal cues spanning only 1–2 frames. This suggests that increasing video encoder complexity does not necessarily improve performance for fast-action detection in $F^3$Set. Notably, simpler models like TSM, paired with advanced 2D CNNs such as RegNet-Y [57], outperform these complex encoders. This highlights the importance of capturing subtle visual differences over short temporal spans, demonstrating that the ability to extract fine-grained temporal cues is more impactful than model complexity.

Head modules such as transformer-based ActionFormer, and GRU-based E2E-Spot, generally outperform other methods. This advantage highlights their effectiveness in capturing long-term temporal dependencies through end-to-end training. Notably, E2E-Spot consistently outperforms ActionFormer across most settings, suggesting that GRU-based architectures may offer an advantageous trade-off between efficiency and representational power for certain types of temporal correlations.

Our proposed $F^3$ED model, leveraging the TSM video encoder, achieves the best performance among all granularity levels. This is attributable to two key design choices: the multi-label classifier and the contextual module. Detailed discussions of these design elements are presented in the next section.

Table 4: Ablation and analysis experiments. The default model takes stride size 2 and clip length 96.

| Experiment | $F^3$Set ($G_{high}$) | | | $F^3$Set ($G_{mid}$) | | | $F^3$Set ($G_{low}$) | | |
|---|---|---|---|---|---|---|---|---|---|
| | $F1_{evt}$ | $F1_{elm}$ | Edit | $F1_{evt}$ | $F1_{elm}$ | Edit | $F1_{evt}$ | $F1_{elm}$ | Edit |
| TSM + E2E-Spot | 31.4 | 71.4 | 68.7 | 39.5 | 72.3 | 77.9 | 60.6 | 78.4 | 82.1 |
| *(a) Feature extractor* | | | | | | | | | |
| I3D [5] (clip-wise) | 22.7 | 59.7 | 68.7 | 27.1 | 60.7 | 74.2 | 51.9 | 67.7 | 78.3 |
| VTN [52] (video transformer) | 14.8 | 58.3 | 56.7 | 20.0 | 59.4 | 68.2 | 39.7 | 63.1 | 73.1 |
| ST-GCN++ [17] (skeleton-based) | 25.4 | 62.1 | 56.1 | 32.4 | 63.9 | 63.5 | 55.1 | 69.4 | 73.2 |
| PoseConv3D [18] ( (skeleton-based)) | 20.1 | 54.5 | 53.2 | 26.0 | 55.4 | 61.9 | 48.8 | 63.0 | 69.7 |
| *(b) Stride size = 4* | 25.9 | 69.2 | 62.7 | 33.4 | 69.9 | 73.0 | 60.0 | 77.9 | 78.8 |
| *Stride size = 8* | 14.0 | 56.7 | 44.3 | 18.5 | 57.4 | 54.8 | 40.4 | 67.0 | 59.2 |
| *(c) without GRU* | 27.6 | 69.0 | 60.6 | 38.0 | 71.3 | 75.3 | 54.7 | 74.1 | 73.4 |
| *(d) Clip Length = 32* | 26.3 | 67.4 | 54.5 | 35.5 | 69.4 | 71.8 | 53.2 | 75.1 | 68.9 |
| *Clip Length = 64* | 30.7 | 71.2 | 67.4 | 38.6 | 72.4 | 77.5 | 58.4 | 77.9 | 81.1 |
| *Clip Length = 192* | 29.3 | 70.3 | 65.7 | 37.3 | 71.4 | 77.0 | 58.8 | 77.1 | 80.4 |
| *(e) Multi-label* | 37.9 | 74.3 | 71.7 | 45.9 | 75.6 | 80.1 | 66.6 | 80.1 | 85.1 |
| *(f) Multi-label + CTX (Transformer)* | 39.0 | 74.3 | 72.8 | 50.5 | 75.5 | 81.8 | 63.4 | 79.6 | 86.8 |
| *Multi-label + CTX (BiGRU)* | 40.3 | 75.2 | 74.0 | 48.0 | 76.5 | 82.4 | 68.4 | 80.0 | 87.2 |

## 5.2 ABLATION STUDY

We selected the highest-performing baseline model (TSM + E2E-Spot) as our default configuration for the subsequent ablation studies. More ablation studies can be found in Appendix H.

**Feature extractor.** An effective feature extractor is crucial for accurate $F^3$ event detection. Below, we summarize some key findings (details in Appendix H). *(1) Frame-wise feature extraction outperforms clip-wise methods*, which divide inputs into non-overlapping segments. Experiments show clip-wise methods produce temporally coarse features and hinder precise event detection. *(2) Transformer-based video encoders* such as VTN [52] struggle on $F^3$Set due to high computational costs and limited ability to effectively capture short-term temporal correlations. *(3) In addition to RGB inputs, we also experimented with skeleton-based pose estimation methods*, including ST-GCN++ [17] and PoseConv3D [18] with human key points as input. While they excel in efficiency and interpretability, they lack critical details like shot direction, limiting performance on $F^3$Set.

**Sparse sampling.** Increasing the stride size allows for a broader temporal coverage within a fixed sequence length. This sparse sampling technique is prevalent in many video understanding tasks [40; 34], offering high efficiency and reasonable accuracy. However, this approach proves inadequate for our task, where events are characterized by their rapid occurrence, frequency, and fine granularity. As illustrated in Table 4(b), increasing the stride size to 4 and 8 leads to a marked decline in performance, underscoring the importance of dense sampling for detecting $F^3$.

**Long-term temporal reasoning.** The default model employs a spatio-temporal video encoder (TSM), complemented by a bidirectional Gated Recurrent Unit [13] (GRU) head for enhanced long-term temporal integration. To assess the necessity of long-term temporal reasoning, we replaced the GRU module with a fully connected layer. The results, presented in Table 4(c), indicate a significant performance decline relative to the original configuration. This finding highlights the essential role of long-term temporal reasoning in analyzing sub-classes such as shot direction, outcomes, and player movements that require information from subsequent frames.

**Clip length.** The sensitivity of sequence models to varying input clip lengths, which encapsulate different temporal contexts, is notable. In $F^3$Set, the incidence of $F^3$ events correlates directly with clip length. Table 4(d) shows that shorter clips result in fewer events per sequence, hindering the model's ability to leverage long-term dependencies among consecutive events effectively. Conversely, while longer clip lengths yield improved results, the marginal gains diminish with increasing length.

**Multi-class versus multi-label classification.** The challenge of modeling over 1,000 possible event type combinations as a multi-class classification problem is formidable. For example, consider two events, $E_1$ (far_ad_bh_stroke_DL_*slice*_apr_in) and $E_2$ (far_ad_bh_stroke_DL_*drop*_apr_in), which differ only in shot technique (*slice* vs. *drop*). Although similar, multi-class classification treats these as distinct classes, thus reducing training efficiency and exacerbating the long-tail distribution bias towards more frequent classes. A more natural approach is multi-label classification, where each event can belong to multiple sub-class elements (e.g., ['far', 'ad', 'serve', 'W', 'in']). Thus, $E1$ and

Table 5: Experimental results on other "semi-$F^3$" datasets.

| Head arch. | ShuttleSet [66] | | FineDiving [69] | | FineGym [58] | | SoccerNetV2 [11] | | CCTV-Pipe [42] | |
|---|---|---|---|---|---|---|---|---|---|---|
| | $F1_{evt}$ | Edit | $F1_{evt}$ | Edit | $F1_{evt}$ | Edit | $F1_{evt}$ | Edit | $F1_{evt}$ | Edit |
| MS-TCN [19] | 70.3 | 74.4 | 65.7 | 92.2 | 57.6 | 65.3 | 43.4 | 74.5 | 25.8 | 31.3 |
| ASformer [71] | 55.9 | 70.6 | 49.9 | 87.6 | 53.6 | 66.3 | 46.3 | 76.1 | 15.4 | 33.4 |
| G-TAD [70] | 48.2 | 61.1 | 52.1 | 82.6 | 45.8 | 51.4 | 42.3 | 72.3 | 31.3 | 33.6 |
| ActionFormer [72] | 62.1 | 67.5 | 68.3 | 92.4 | 54.0 | 59.7 | 43.0 | 64.6 | 18.8 | 29.5 |
| E2E-Spot [24] | 70.2 | 75.0 | 75.8 | 93.7 | 62.1 | 65.4 | 46.2 | 72.9 | 27.2 | 35.2 |
| $F^3$ED | 70.7 | 77.1 | 77.6 | 95.1 | 70.9 | 70.7 | 48.1 | 76.6 | 37.0 | 39.5 |

$E2$ only differ in shot technique but are identical in other aspects. This adjustment facilitates more effective training and shows an increase in performance, as shown in Table 4(e).

**Contextual knowledge.** Beyond the statistical results in Table 3, analysis of predicted event sequences reveals that current baselines may produce invalid sequences due to logical errors or uncommon practices. For instance, a right-handed player cannot logically direct a forehand shot from the deuce court as "II" or "IO". Similarly, an event ending in a winner or error should logically conclude the sequence. Additionally, it is uncommon for a player to hit with backhand when the ball is played to their forehand side. Further examples are detailed in Appendix I. These observations indicate that existing baselines fail to effectively capture event-wise contextual correlations. By adding the CTX module, the performance further increases as shown in Table 4(f). We also compared BiGRU and Transformer Encoder for the CTX module. BiGRU performed slightly better, likely due to its efficiency in modeling short *event sequences* (usually $< 20$ per clip) with fewer parameters.

## 5.3 GENERALIZABILITY TO "SEMI-$F^3$" DATA

$F^3$ task possesses broad applicability across numerous real-world domains, such as sports, autonomous driving, surveillance, and production line inspection. Nevertheless, creating such a $F^3$ dataset necessitates substantial expertise and extensive labeling efforts. We have found that existing video datasets often fail to fully address all three dimensions of the $F^3$ task—"fast", "frequent", and "fine-grained". In this section, we conducted experiments on several "semi-$F^3$" datasets that partially meet these criteria, including Shuttleset [66] for badminton (racket sport), FineDiving [69] for diving (individual sports), FineGym [58] for gymnastics (individual sports), SoccerNetV2 [50] (team sports), and CCTV-Pipe [42] for pipe defect detection (industrial application). We report only the $F1_{evt}$ and Edit score, as not all datasets necessitate multi-label classification given their limited event types. For the video encoder, we chose TSM, which consistently outperforms the others on average.

Performance across different domains can vary significantly depending on the difficulty of tasks and the scale of datasets. For instance, the CCTV-Pipe dataset, targeting temporal defect localization in urban pipe systems, shows suboptimal performance due to factors such as ambiguous single-frame annotations for each defect, multiple defects at the same time, long-tailed distribution of defect types, and limited dataset size. Our performance is better than the results reported in [42]. Generally, methods that effectively handle $F^3$Set tend to perform well across other applications, as indicated in Table 5. Our $F^3$ED outperforms existing baselines in all datasets, demonstrating its robust generalizability for detecting "semi-$F^3$" events across various domains. While $F^3$ event detection benefits from accurate event localization, a high-performing LCL module is not a hard prerequisite (see Appendix J). Therefore, our method can be generalized and benefit broader applications.

## 6 CONCLUSION AND FUTURE WORK

In this study, we addressed the challenge of analyzing fast, frequent, and fine-grained ($F^3$) events from videos by introducing $F^3$Set, a benchmark for precise temporal $F^3$ event detection. $F^3$Set datasets usually feature detailed event types (approximately 1,000), annotated with precise timestamps, and provide multi-level granularity. We have also developed a general annotation toolchain that enables domain experts to create $F^3$ datasets, thereby facilitating further research in this field. Moreover, we proposed $F^3$ED, an end-to-end model that effectively detects complex event sequences from videos, using a combination of visual features and contextual sequence refinement. Our comprehensive evaluations and ablation studies of leading methods in temporal action understanding on $F^3$Set highlighted their performance and provided critical insights into their capabilities and limitations. Moving forward, we aim to extend the scope of $F^3$ task to more real-world scenarios and advance the development of $F^3$ video understanding.

ACKNOWLEDGMENTS

This research is supported by the AI Singapore (AISG3-RP-2022-030). Any opinions, findings and conclusions or recommendations expressed in this material are those of the author(s) and do not reflect the views of funding bodies.

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
