# OpenReview forum: "$F^3Set$: Towards Analyzing Fast, Frequent, and Fine-grained Events from Videos"
_ICLR.cc/2025/Conference — ICLR 2025 Poster_

### Official Review · Reviewer_YGio · 2024-10-18

**Soundness:** 2
**Presentation:** 3
**Contribution:** 3
**Rating:** 8
**Confidence:** 4

**Summary:**

The paper introduces **F3Set**, a novel sports-related dataset designed to address the challenges of detecting fast, frequent, and fine-grained (F3) events from videos. The dataset contains over *1,000* event types annotated with precise timestamps, primarily focused on tennis, but also includes other sports such as badminton and table tennis, with the potential to be extended to various other sports. To tackle the challenges of event classification and localization, the authors propose **F3ED**, a model that utilizes a video encoder for spatial-temporal feature extraction, a multi-label event classifier, and a contextual module to refine event sequence predictions. The model is evaluated on F3Set and demonstrates superior performance over certain models in both event localization and classification. Additionally, the authors provide a semi-automated toolchain for annotating F3 events, making the dataset scalable for use in other sports.

**Strengths:**

- **Originality**: The paper introduces a dataset designed for detecting **fast, frequent, and fine-grained events** in sports videos. While there have been other tennis-related datasets, such as **Tennis Stroke Dataset** and **TennisDB**, **F3Set** stands out due to its **1,000+ annotated event types** and precise annotations with fine-grained details.

- **Quality**: The dataset contains over **11,584 video clips** of professional tennis matches, featuring **42,846 tennis shots** and detailed annotations. Each shot is labeled with attributes like shot type, direction, and outcome, allowing for comprehensive event analysis. Also dataset has been collected in high resolution and moderate fps 25-30, which can be utilized for tasks beyond event classification, such as pose estimation and movement analysis.

- **Clarity**: The authors provide clear examples of how annotations are done, such as distinguishing between forehand and backhand shots, and detailing shot outcomes like winners or errors. The explanations of **multi-level granularity** make it easy to understand how the dataset can be applied to different tasks, from coarse to fine event classification. In addition, the authors did a very good job providing the explanation of tennis terminology.


- **Code and Tools**: The authors provide a **well-structured and anonymized codebase**, along with a semi-automated toolchain for efficient event annotation.  This makes it easier for researchers to adopt the dataset and extend it to other domains.

- **Dataset Diversity**: The dataset includes **high-resolution videos** from 114 professional tennis matches featuring both men and women players, with frame rates of 25-30 FPS. This diversity, along with specific annotations for both right- and left-handed players, ensures the dataset can support a wide range of analyses.

**Weaknesses:**

- **Significance**: While the dataset is beneficial for sports like tennis, badminton, and table tennis, the methodology is highly limited due to the nature of these sports. These sports are relatively easier to model with their controlled environments and predictable movement patterns, but the methodology may not generalize well to faster and more complex sports like **soccer** or **basketball**, which require higher FPS rates and need to account for multi-player interactions.

- **Benchmark Simplicity**: The proposed benchmark, while interesting and comprehensive in its approach, is relatively simple. The choice to crop the input videos to **224x224 resolution** while originally collecting them in higher resolution raises questions. The authors claim that F3ED outperforms more complex models like **SlowFast**, but this might be due to the limited resolution of the input images, which could fail to capture subtle visual distinctions. More work is needed to confirm or debunk this claim, including testing higher-resolution inputs to better understand the effects of image quality on model performance.


- **Dataset Nature**: While the dataset is relatively diverse, there are still questions regarding the impact of **camera angles**, **court types**, **weather conditions**, and **illumination**. In real-world settings, these factors can vary significantly and may affect model performance. While in professional competitions these variables might be more consistent, in practical scenarios such variations could play an important role in the robustness of event detection. Ideally, dataset and benchmark should thoroughly address those concerns.

- **Fit with ICLR and Representation Learning**: A notable weakness of this paper is that it does not explicitly address how the proposed model learns **representations** or how these representations could be generalized or transferred to other domains. Additionally, the paper primarily compares its performance to models with similar architectural structures, such as 3D CNNs, without exploring **fundamentally different approaches** to event detection. For instance, instead of relying on crops, the authors could have explored using **pose estimation techniques** to detect human poses and tackle the problem from a different perspective. Comparing such an approach across metrics like accuracy and speed, and then resonating on why one method outperforms or underperforms the other, would have provided valuable insights into the advantages or limitations of the proposed approach.

**Questions:**

- **Generalization of Methodology to Other Sports**: How do you expect the proposed methodology to generalize to sports with faster movements and multiple players, such as **soccer** or **basketball**? Can the model handle the requirements for higher frame rates and multi-human event interactions, or are there adjustments needed for such cases?

- **Use of Low-Resolution Input (224x224)**: The decision to use **224x224 crops** despite collecting higher resolution videos raises questions. Have you tested the model on **higher resolution inputs**, and if so, how did it perform? Could higher resolution provide more visual details and improve the model’s ability to detect subtle distinctions?

- **Alternative Approaches for Event Detection**: The paper focuses on comparing the proposed method with models using **similar architectural foundations** like 3D CNNs. Have you considered comparing your method to **fundamentally different approaches**, such as **pose estimation** for event detection (In other words: extracting poses and then utilizing coordinates to classify events)? How would such comparisons impact performance in terms of accuracy, speed, and interpretability?

- **Dataset and Camera Perspectives**: The dataset appears to focus on controlled environments, but real-world scenarios often involve **different camera angles, lighting conditions, and varying court types**. How would your model perform under different camera perspectives or in less controlled environments? Have you tested the model with variations in lighting or weather conditions?

- **Scalability of Annotation Process**: The paper mentions a **semi-automated toolchain** for annotating events. How scalable is this annotation process, especially if applied to larger datasets or more complex sports with multiple participants?

---

> ### Author Response · Authors · 2024-11-20
> **Rebuttal by Authors**
>
> Dear reviewer,
>
> We appreciate your valuable feedback and comments. We have carefully considered the comments and have made the following responses:
>
> **Q1: Generalization of Methodology to Other Sports**
>
> A1: Thank you for raising this question. We acknowledge the concerns regarding the generalizability of our methodology to sports with faster movements and multiple players, such as soccer or basketball. To address this, we conducted additional experiments on SoccerNetV2 [1], a dataset designed for soccer video understanding. We focus on the "ball action spotting" [2] task which involves identifying the timing and type of ball-related actions across 12 classes (e.g., Pass, Shot, Cross), with each action annotated by a single timestamp. As shown in the table below, our method, F3ED, achieves superior performance compared to baseline methods, further validating its generalizability to faster, multi-player sports.
>
> |video encoder|head arch.|F1_evt|Edit|
> |-|-|-|-|
> |TSM|TCN|39.8|74.0|
> ||MSTCN|43.4|74.5|
> ||Asformer|46.3|76.1|
> ||G-TAD|42.3|72.3|
> ||ActionFormer|43.0|64.6|
> ||E2E-Spot|46.2|72.9|
> ||F3ED|48.1|76.6|
>
> For basketball, we observed that most existing datasets focus on action recognition rather than temporal action detection, making it hard to test our method. We will include the results on SoccerNetV2 in the revised manuscript to show the generalizability of our method.
>
> [1] "Soccernet-v2: A dataset and benchmarks for holistic understanding of broadcast soccer videos." CVPR 2021.
> [2] https://www.soccer-net.org/tasks/ball-action-spotting
>
> **Q2: Use of Low-Resolution Input (224x224)**
>
> A2: Thank you for your value suggestion. We understand the concern regarding low resolution inputs (224x224). To address this, we conducted additional experiments to analyze the effects of using higher-resolution inputs on model performance. The results are summarized in the table below.
>
> First, we evaluated F3ED using TSM as the video encoder with input resolutions of 224x224, 336x336, and 448x448. The results show a consistent improvement in performance as resolution increases, though the gains diminish at higher resolutions (e.g., 448x448). This suggests that while higher resolutions can provide additional visual details, the marginal benefits decrease beyond a certain point.
>
> We also tested F3ED with SlowFast as encoder on higher-resolution inputs (336x336). We experimented with two variants: SlowFast 4x16 and SlowFast 8x8. Despite higher complexity, SlowFast 4x16 underperformed TSM, likely due to its lower temporal resolution, which limits its ability to capture subtle differences. SlowFast 8x8 achieved slightly better performance than SlowFast 4x16 and marginally outperformed TSM in Edit score (+0.2) but lagged in F1_evt and F1_elm metrics.
>
> The 224x224 resolution remains a common choice in video analytics due to its efficiency and compatibility with pre-trained models. Balancing complexity, performance, and efficiency, we selected 224x224 and TSM as the default configuration for F3ED. We will include the above analysis in the revised manuscript to clarify the trade-offs between resolution, complexity, and performance.
>
> |Video encoder|Head arch.|resolution|params(M)|FLOPs|F3Set (Ghigh)|||
> |-|-|-|-|-|-|-|-|
> ||||||F1_evt|F1_elm|Edit|F1_evt|F1_elm|Edit|F1_evt|F1_elm|Edit|
> |TSM|F3ED|224x224|5.6|77.3|40.3|75.2|74.0|
> |TSM|F3ED|336x336|5.6|77.3|43.2|77.1|74.8|
> |TSM|F3ED|448x448|5.6|77.3|44.4|78.1|74.5|
> |SlowFast 4x16|F3ED|336x336|52.7|494.2|37.4|73.7|71.3|
> |SlowFast 8x8|F3ED|336x336|52.8|903.7|41.0|74.6|75.0|

---

> ### Author Response · Authors · 2024-11-20
> **Rebuttal by Authors**
>
> **Q3: Dataset Nature and Camera Perspectives**
>
> A3: Thank you for your insightful feedback. We recognize the importance of testing generalizability under diverse real-world conditions, including variations in camera angles, court types, weather, and illumination. While professional competition videos are filmed under relatively standardized conditions, we have ensured that F3Set captures a significant level of diversity in these factors:
> - Camera Angles: The dataset includes videos from 114 broadcast matches across various tournaments, each exhibiting different camera angles. Additionally, individual matches often feature multiple perspectives: standard bird’s-eye view and low-angle view shots from behind one player, where the foreground player appears significantly larger than the background player.
> - Court Types: The dataset covers all three tennis court surfaces—hard court, clay court, and grass court—with diverse color schemes (e.g. blue, green, red, black, green, purple, etc).
> - Weather Conditions and Illumination: The videos in F3Set reflect diverse weather and lighting scenarios, including day and night matches, cloudy weather, indoor and outdoor games, and challenging conditions such as partial sunlight casting shadows on the court. For outdoor matches, some videos feature strong sunlight on parts of the court, making it harder to track the ball or players in those areas. Indoor matches also vary in brightness, with some tournaments having brighter lighting setups and others being relatively dimmer.
>
> To further enhance the diversity, we have collected videos from college-level matches for tennis doubles [3]. We are also actively expanding F3Set by incorporating more matches from platforms like UTR [4] and junior-level competitions using our annotation toolchain.
>
> [3] https://github.com/F3Set/F3Set/tree/main/data/f3set-tennis-doubles
> [4] https://www.utrsports.net/
>
> **Q3: Alternative Approaches for Event Detection**
>
> Thank you for your valuable suggestions. We recognize the potential of using human pose estimation for representation learning and its ability to generalize to other domains. To explore this, we conducted additional experiments leveraging skeleton-based representations for event classification in F3Set. We used MMPose to extract player keypoints from 1280x720 resolution images, generating skeleton data. Two advanced skeleton feature extractors ST-GCN++ [5] (GCN-based) and PoseConv3D [6] (CNN-based) were evaluated. The extracted skeleton features were processed with the F3ED head architecture for classification and localization. The results are summarized in the table below.
>
> |Feature extractor|Head arch.|Params (M)|Inference time per frame (ms)|F3Set (Ghigh)|||F3Set (Gmid)|||F3Set (Glow)|||
> |-|-|-|-|-|-|-|-|-|-|-|-|-|
> |||||F1_evt|F1_elm|Edit|F1_evt|F1_elm|Edit|F1_evt|F1_elm|Edit|
> |TSM|F3ED|5.6|10.6|40.3|75.2|74.0|48.0|76.5|82.4|68.4|80.0|87.2|
> |ST-GCN++ [5]|F3ED|2.3|4.0|25.4|62.1|56.1|32.4|63.9|63.5|55.1|69.4|73.2|
> |PoseConv3D [6]|F3ED|6.8|6.4|20.1|54.5|53.2|26.0|55.4|61.9|48.8|63.0|69.7|
>
> Key findings include:
> 1. Among the two skeleton-based methods, ST-GCN++ demonstrated better overall performance.
> 2. Visual features extracted from RGB images using TSM consistently outperformed skeleton-based methods in all three granularities. This is likely because many event types in F3Set include information such as shot direction and shot outcomes, which skeletal data cannot capture.
> 3. Skeleton-based methods excel in computational efficiency and interpretability, requiring fewer parameters and offering faster inference, while directly highlighting player movements and poses.
>
> While skeleton-based approaches may not fully match the performance of RGB-based models for F3Set, they offer unique benefits, particularly in terms of speed and transparency. We plan to further investigate skeleton-based methods and their integration with visual features in future work.
>
> [5] "PYSKL: Towards Good Practices for Skeleton Action Recognition," ACMMM 2022.
> [6] "Revisiting Skeleton-based Action Recognition," CVPR 2022.
>
> **Q5: Scalability of Annotation Process**
>
> Thank you for your question regarding the scalability of our semi-automated annotation toolchain.
>
> For sports with multiple participants, our toolchain has been successfully applied to annotate tennis doubles matches, which involve four players on the court simultaneously. It supports player detection, enabling users to track players of interest and specify who performed each action.
>
> For scalability, our toolchain incorporates several automated processes as discussed in Section 3. To be clearer, annotating a 2-hour tennis singles match typically requires about 2.5 hours of manual effort. To further enhance scalability, we can integrate our event detection model into the annotation pipeline. The model is iteratively trained as annotations are added, reducing manual work as accuracy improves. This enables scalability to larger datasets.

---

> ### Author Response · Authors · 2024-11-22
> **A gentle reminder for discussion**
>
> Dear Reviewer YGio,
>
> Thank you for the time and effort you have invested in reviewing our paper and providing insightful feedback. We have carefully addressed your concerns and uploaded a revised manuscript, with all modifications highlighted in blue. Due to page limitations, we have also included additional details in the Appendix, similarly highlighted.
>
> We would be grateful if you could review our revisions and rebuttal. Should you require further clarification or additional details on any points, we are happy to provide more information. Thank you once again for your thoughtful consideration.
>
> Best regards,
> The Authors

---

> ### Author Response · Authors · 2024-11-24
> **A gentle reminder for discussion**
>
> Dear Reviewer YGio,
>
> Thank you once again for your valuable time and effort in reviewing our work and providing insightful feedback. We would be happy to discuss any further questions and comments you may have. Please let us know if you have received our responses and if we have successfully addressed your concerns. We would appreciate your prompt response. Thank you once again for your feedback.
>
> Best regards,
>
> The Authors

---

> > ### Author Response · Authors · 2024-11-26
> > **Follow-up on Our Response and Revised Manuscript**
> >
> > Dear Reviewer YGio,
> >
> > We wanted to kindly follow up to inquire whether our response and the revised manuscript have adequately addressed your concerns. As the deadline for uploading the final revised PDF approaches, we would greatly appreciate knowing if you have any further suggestions or feedback on our revisions. Thank you again for your time and thoughtful review. We look forward to hearing from you.
> >
> > Best regards,
> >
> > The Authors

---

> > > ### Comment · Reviewer_YGio · 2024-11-26
> > > **Subsntantial clariicatoin made, Thanks!**
> > >
> > > Thank you for addressing the concerns I raised in my previous comments. After reviewing your responses and reading the updated manuscript, I believe my concerns have been adequately addressed. I look forward to seeing how the presented work progresses, particularly with the incorporation of results involving skeletons.
> > >
> > > Given your comprehensive responses and the adjustments made to the paper, I am happy to increase my score by 1. Regarding the ethical considerations, since the dataset is licensed under CC BY 4.0 International, I  don't think there are any ethical issues to be addressed.
> > >
> > > I look forward to following your work and wish you great success in this endeavor.

---

> > > > ### Author Response · Authors · 2024-11-27
> > > >
> > > > Dear Reviewer YGio,
> > > >
> > > > Thank you for your thoughtful feedback and for raising the score. We sincerely appreciate your recognition of our efforts and your constructive suggestions. We will ensure to address your suggestions, particularly the incorporation of results involving skeletons, in the final version. Thank you again for your encouragement and support!
> > > >
> > > > Best regards,
> > > >
> > > > The Authors

---

### Official Review · Reviewer_bJ2T · 2024-10-28

**Soundness:** 3
**Presentation:** 1
**Contribution:** 2
**Rating:** 5
**Confidence:** 5

**Summary:**

This paper mainly introduces a dataset for advancing fast, frequent and fine grained video processing tasks.

They also introduce a model built on top of the newly introduced dataset, and show improvements over several variants of existing video encoders and temporal fusion modules, eg, transformer-based modules, etc.

They choose some video encoders such as SlowFast, TSN and TSM, with different head architectures and compared to their proposed model. Experiments on 3 levels of granularity show the effectiveness of the model.

They also evaluate their module on top of TSM video encoder, on some fine grained benchmarks and show consistent improvements.

**Strengths:**

+ The efforts of collecting a dataset for fast, frequent and fine grained events are appreciated.

+ The authors identify an interesting research problem, in terms of fast, frequent and fine grained video-related research.

+ Table 1 is a nice and interesting comparisons.

+ Some interesting comparisons and experiments conducted, and the authors also provide some valuable insights, especially section 5.1.

**Weaknesses:**

Major:

- The dataset currently focuses on tennis swing, the contribution is limited (i) how your dataset differs from existing dataset and (ii) the dataset at this stage is only tennis, what would be the scope and focus of this benchmark.

- The paper is written in a rush, and not well-structured. The authors mention that “our benchmark can contribute towards a VideoNet for future LLM’s benchmarking and finetuning”, but where and how? The contributions listed are also a bit vague and weak at this stage.

- A review of existing closely related works for the concept of “fast, frequent and fine grained” is not performed; however, the paper introduces the new model (i) any existing works in the literature and how the proposed method differs from existing works (ii) any insights from reviewing existing works, eg, any practical concerns that highlight the importance of introducing a new model, etc.

- Sec. 3 is not well-written and well-structured. The reviewer suggests the authors refine this work, and make it solid enough for next venue. Having a complete understanding, review and analysis in this work, would be much appreciated.

- The justification of choosing evaluation metrics is not provided, and how these evaluation metrics contribute to the evaluation of performance etc. Section 5 page 7 bottom, what does it mean by “we have adapted these methods to develop …”, what methods and how this is performed?

- Regarding evaluations, in Table 2, the use of video encoders is a bit too limited, although the reviewer likes the interesting comparisons presented. The authors should refine their research aims and make it better and clear. The current scope a bit too big, and also core evacuations a bit too limited. This would affect the impact of the work.

- Showing the evaluation results in the form of list is not good, eg, in section 5.1. More detailed discussions are indeed needed to show more in depth analysis and comparisons. Although the paper discusses some insights, the paper at this stage is too raw and requires further efforts to make it solid and thorough enough. Last sentence of section 5.1 does not provide much information. “… achieves optimal performance among all methods”, how and why?

Minor:

- The fine grained concept should be first introduced in the introduction section, eg what is fine grained recognition tasks in videos, and what is the concept of frequent and fast.

- All the experimental results and evaluations are presented in the form of tables. The authors are encouraged to use some plots, visualisations to make the comparisons clear and vivid to researchers and readers.

- As a research paper, it is suggested to make it as clear as possible, eg “Conflicting samples were resolved using a majority vote criterion”, but how?

- A notation section detailing the maths symbols used in the paper would be better.

- Section 4, the proposed model is also unclear to reviewer. Inside problem formulation, what is that “j”? Fig. 4 is also not clear to reviewer. The figure caption does not provide much information to understand the figure.

- In Introduction section, what is “T”, body and wide? These concepts are unclear to reviewer. The concepts are not very well-explained.

- Incomplete sentence in sec. 3.1, “If it lands in bounds after crossing the net.”, it would be better to have a figure explain these concepts clearly in lexicon section would help readers. Fig. 3 suffers from low resolution, and the fonts are unable to read due to its too small nature.

**Questions:**

Please refer to my concerns above.

**Details Of Ethics Concerns:**

- Copyright issues.

- Human subjects related research.

- Potential ethics issues.

- Dataset biases issues regarding locations, nationalities, etc.

- A model eg trained on a biased dataset, would that cause potential issues in recommendations. How would these concerns being addressed etc.

---

> ### Author Response · Authors · 2024-11-20
> **Rebuttal by Authors**
>
> Dear reviewer,
>
> We appreciate your feedback and comments and have carefully considered them to make the following responses:
>
> **Q1: The dataset currently focuses on tennis swing, the contribution is limited (i) how your dataset differs from existing dataset and (ii) the dataset at this stage is only tennis, what would be the scope and focus of this benchmark.**
>
> A1: Thank you for your feedback. We would like to clarify that the F3Set dataset we proposed encompasses multiple sports datasets, including tennis singles and doubles matches, badminton, and table tennis, rather than solely focusing on tennis swings. This broader scope demonstrates the flexibility and potential applications of our dataset framework beyond a single sport. Additionally, our framework includes a general annotation toolchain, enabling domain experts to create new F3 datasets for various applications. These aspects are highlighted in the Abstract, Section 1, and at the beginning of Section 3.
>
> To address the differences between F3Set and existing datasets, we provide a comprehensive comparison of our dataset with existing F3-related datasets in Section 2 (L137–L152). This includes a summary of key differences in terms of task, scale, and the unique characteristics of fast, frequent, and fine-grained tasks. Table 1 also provides a statistical summary, highlighting the distinct features of F3Set compared to other related datasets. We will clarify these claims and highlight the broader scope and key differences of our dataset in the revised manuscript.
>
> **Q2: The paper is written in a rush, and not well-structured. The authors mention that “our benchmark can contribute towards a VideoNet for future LLM’s benchmarking and finetuning”, but where and how? The contributions listed are also a bit vague and weak at this stage.**
>
> A2: Thank you for your feedback. We acknowledge that the phrasing of certain claims, such as "our benchmark can contribute towards a VideoNet for future LLM’s benchmarking and fine-tuning," could have been more precise. As noted in the fourth paragraph (L077–L085) of the introduction, current multi-modal LLMs, while effective in some areas, exhibit limitations in understanding fast, frequent, and fine-grained (F3) events, as highlighted by the preliminary experiments in Appendix A. Therefore, by introducing F3Set, we hope it can help advance multi-modal LLM capabilities in F3 video understanding, which is a potential future direction.
>
> Regarding the contributions of our paper, we apologize if they appeared unclear or weak. To clarify, the core contributions of our work include the following:
>
> - We introduce F3Set, a new benchmark with datasets covering multiple sports addressing the fast, frequent, and fine-grained event detection tasks, distinguishing it from existing datasets.
> - We propose a general annotation toolchain that facilitates the extension of F3Set to new domains and applications.
> - As F3Set falls within the domain of temporal action understanding, we conduct comprehensive evaluations and ablation studies using leading methods in this field. We also provide a detailed analysis of the results to offer valuable insights for future research.
> - Recognizing the limitations of existing approaches, we propose F3ED, an end-to-end model designed to accurately detect F3 events. F3ED outperforms existing baselines and can be used for future benchmarking and development.
>
> These contributions are outlined in the introduction and substantiated through our experiments and analyses. We would appreciate further clarification on which aspects of our contributions you found vague or weak. This feedback will help us focus our efforts to better address your concerns.

---

> ### Author Response · Authors · 2024-11-20
> **Rebuttal by Authors**
>
> **Q3: How does the proposed method differ from existing works, and what insights highlight the need for a new model?**
>
> A3: Thank you for raising this point. We have discussed existing works in both Section 1 (L067–L076) and Section 2 (L104–L136), but we acknowledge that this could be made more explicit.
>
> To summarize, F3Set targets the detection of fast, frequent, and fine-grained (F3) events with precise temporal annotations. This task is related to fine-grained action recognition, temporal action localization (TAL), temporal action segmentation (TAS), and temporal action spotting (TASpot). However, we observe limitations in how these methods address the specific challenges of F3 event detection. For instance, fine-grained action recognition often assigns a single label to an entire video, which fails to capture sequences of events occurring at precise moments. Recent TAL and TAS methods often rely on pre-trained or modestly fine-tuned input features, which are insufficient to capture the domain-specific visual subtleties required for diverse F3 events. TASpot approaches, despite their use of dense frame sampling and end-to-end training, often result in sparse temporal predictions, making it difficult to capture event-wise causal relationships effectively, particularly when the number of action types increases.
>
> Our proposed F3ED method addresses these gaps through the following designs:
> - Frame-wise dense processing and end-to-end training: F3ED enables efficient training and inference for frame-wise spatial-temporal feature extraction using modern CNNs with TSM, tailored for densely sampled frames.
> - Multi-label classification: By replacing the traditional multi-class event classifier with a multi-label classifier, F3ED improves training efficiency and better handles events with minor differences (e.g., subtle distinctions in shot techniques while other attributes remain the same).
> - Contextual module: F3ED incorporates a contextual module to capture event-wise causal correlations and refine predictions by leveraging logical patterns within event sequences, reducing errors and accounting for uncommon practices that may be challenging to infer from visual clues alone.
>
> By addressing these limitations, F3ED offers a practical and robust solution for F3 event detection. We will improve the "Related Work" section to provide a more comprehensive and explicit review of existing works and their limitations.
>
> **Q4: Sec. 3 is not well-written and well-structured. The reviewer suggests the authors refine this work, and make it solid enough for next venue. Having a complete understanding, review and analysis in this work, would be much appreciated.**
>
> A4: Thank you for your feedback. We would like to clarify the structure and logic of this section as follows:
> - Overview: We provide an introduction to this section, outlining the components of F3Set to be covered, including the definition of F3 events, the dataset construction process, and its statistics and properties.
> - Event Description: We formally define the F3 events and their properties in F3Set using tennis singles matches as an example.
> - Dataset Creation Process: We detail the end-to-end pipeline for creating a F3Set dataset, which includes data collection and data annotation process.
> - Dataset Statistics and Properties: We present the statistics and properties of a F3Set dataset.
>
> To illustrate the process, we use tennis singles as a case study. The same methodology has been applied to other sports, including table tennis, badminton, and tennis doubles, demonstrating the extensibility and generalizability of our approach.
>
> We believe Section 3 is logically structured. We would greatly appreciate it if the reviewer could point out specific parts of Section 3 that they found unclear or insufficiently detailed. This feedback will help us focus our revisions to enhance the clarity and robustness of this section.

---

> ### Author Response · Authors · 2024-11-20
> **Rebuttal by Authors**
>
> **Q5: What is the justification for the chosen evaluation metrics, and what methods were adapted in Section 5, page 7? How were they modified?**
>
> A5: Thank you for raising these important points. The choice of evaluation metrics is discussed in Section 5 (L363–L371). To summarize,
> edit score measures similarity between predicted and ground truth sequences using Levenshtein distance; mean F1 with temporal tolerance assesses classification and localization accuracy within ±δ frame. These metrics align with evaluation standards in similar tasks [1, 2]. For example, [1] uses edit score and mean F1 for table tennis stroke recognition, evaluating F1 at overlap thresholds for temporally segmented actions. Similarly, [2] identifies precise event timestamps using Average Precision within ±δ frames, equivalent to our F1_evt. To support multi-label classification in our task, we introduced F1_elm with ±δ frames for evaluating multiple labels.
>
> This refers to modifications made to temporal action understanding methods (e.g., TAL, TAS) to suit F3 event detection. Direct application of these methods often yields suboptimal results due to: 1) some are not designed for end-to-end training, and reliance on pre-trained video extractors leads to poor performance in domain-specific scenarios; 2) some methods process videos in non-overlapping segments rather than on a per-frame basis, making it difficult to identify precise event timestamps.
>
> We addressed these issues by: 1) enabling end-to-end training for domain-specific fine-tuning; 2) supporting frame-wise dense predictions for precise timestamping. These adaptations ensure fair comparisons with our method while retaining the essence of the original approaches.
>
> [1] "ViSTec: Video Modeling for Sports Technique Recognition and Tactical Analysis." AAAI 2024.
> [2] Hong, James, et al. "Spotting temporally precise, fine-grained events in video." ECCV 2022.
>
> **Q6: The use of video encoders in Table 2 is limited, and the scope and core evaluations need refinement for clarity and impact.**
>
> A6: Thank you for your feedback on the scope of video encoders. We balanced extraction capability and model complexity to enable efficient end-to-end processing of long-frame sequences without exceeding memory limits. While advanced transformer-based encoders like TimeSformer [3] and VideoSwin [4] excel in many tasks, their complexity often prevents end-to-end processing of long sequences.
>
> To address your concern, we tested VTN [6], a transformer-based encoder supporting frame-wise extraction. However, due to its computational cost, we could only process 64-frame sequences with a batch size of 3, achieving suboptimal results on F3Set (G_high): 55.7 (edit score) and 50.0 (mean F1_elm), significantly lower than other encoders. This may be due to VTN's limited ability to capture short-term temporal correlations and subtle visual differences—key for detecting F3 events.
>
> We will include this evaluation in the revised manuscript to provide a more complete understanding of encoder choices and performance.
>
> [3] "Is space-time attention all you need for video understanding?." ICML 2021.
> [4] "Swin transformer v2: Scaling up capacity and resolution." CVPR 2022.
> [5] "Video transformer network." ICCV 2021.
>
> **Q7: The evaluation results in Section 5.1 need more detailed discussions and in-depth analysis, as the current list format and concluding statement lack clarity and depth.**
>
> A7: Thank you for your suggestions regarding Section 5.1. We will restructure this section into paragraphs to create a more cohesive and reader-friendly narrative. As for the statement, “… achieves optimal performance among all methods,” a detailed explanation is provided in the subsequent Ablation Study section, where we analyze the contributions of the multi-label classifier and contextual module. We will clarify this connection in Section 5.1 to enhance clarity.
>
> **Q8: Introduce the concepts of fine-grained, frequent, and fast recognition tasks in the introduction.**
>
> A8: Thank you for your feedback. The concepts of "fast," "frequent," and "fine-grained" are briefly introduced in the first sentence of the Introduction, where we define them as recognizing sequences of fast-paced events (fast), involving many actions in a short period (frequent), and diverse event types (fine-grained) with precise timestamps (±1–2 frames). To further elaborate, we also provide a concrete example using tennis events in the third paragraph of the Introduction (L048–L076), use Figure 1 for demonstration. However, we acknowledge that terms like "fine-grained" may lack universally agreed definitions. To address this, we quantitatively analyze and compare F3Set with existing F3 related datasets in terms of number of event types (fine-grained), event duration (fast), and number of events per second (frequent) in Table 1. This demonstrates that F3Set comprehensively addresses all F3 aspects, whereas other datasets do not fully cover these dimensions.

---

> ### Author Response · Authors · 2024-11-20
> **Rebuttal by Authors**
>
> **Q9: Replace tables with plots or visualizations for clearer comparisons.**
>
> A9: Thank you for your suggestion. We agree that incorporating plots and visualizations can make comparisons clearer and more engaging. To address this, we will convert some of the tables, such as Table 4, into plots to improve the presentation and enhance the clarity of our results.
>
> **Q10: As a research paper, it is suggested to make it as clear as possible, eg “Conflicting samples were resolved using a majority vote criterion”, but how?**
>
> A10: Thank you for your feedback. To clarify, we conducted multiple rounds of cross-validation, which involved randomly sampling rallies and performing quality checks among annotators to ensure the accuracy of event-based labels. In cases where conflicting annotations arose, annotators were asked to input the labels they believed to be correct. The final label for each event was determined based on a majority vote among the annotators. This process ensured that the final annotations were robust and reflected a consensus among domain experts.
>
> **Q11-12: Provide a notation section for math symbols and clarify the proposed model, including the meaning of “j” in problem formulation and improving Figure 4 and its caption.**
>
> A11-12: Thank you for your feedback. The symbol "j" represents the j-th element in an event. We define the number of elements per event (discussed in Section 3.1) to be K. Therefore, "j" will be an integer in {1, 2, ..., K}. We have provided explanations of the mathematical symbols used in Section 4, "Problem Formulation". We can add a notation section summarizing all mathematical symbols if you think it will be more clear.
>
> Additionally, we will revise the caption for Fig. 4 to provide more detailed information by saying "The F3ED architecture: (1) a video encoder for spatial-temporal feature extraction, (2) an event detector to pinpoint critical event timestamps, (3) a multi-label event classifier leveraging visual features at the identified timestamps, and (4) a contextual module that refines predictions using sequence-level contextual knowledge. The ‘plus’ symbol combines outputs from the localizer (LCL) and classifier (MLC), resulting in a sequence of multi-label event representations passed to the CTX module. Red squares indicate errors (e.g., incorrect elements) in the multi-label representation due to purely visual predictions."
>
> **Q13: In Introduction section, what is “T”, body and wide? These concepts are unclear to reviewer. The concepts are not very well-explained.**
>
> A13: Thank you for your feedback. "T," "Body," and "Wide" refer to tennis serve directions and are explained in detail in Appendix B. We will add a note in the Introduction directing readers to Appendix B for clarity.
>
> **Q14: Section 3.1 has an incomplete sentence, and Fig. 3 needs higher resolution and clearer fonts for better readability.**
>
> A14: Thank you for noting the typo in Section 3.1. The corrected sentence is: “The shot taken by the receiver after a serve is called a ‘return’ if it lands in bounds after crossing the net.” We will also replaced Figure 3 with a higher-resolution version and enlarged the fonts for better readability.
>
> **Q15: Details Of Ethics Concerns: Copyright issues. Human subjects related research. Potential ethics issues. Dataset biases issues regarding locations, nationalities, etc. A model eg trained on a biased dataset, would that cause potential issues in recommendations. How would these concerns being addressed etc.**
>
> A15: Thank you for raising this point. We believe our dataset and methodology adhere to ethical standards, and we have not identified any specific ethical issues related to privacy, security, discrimination, or fairness in our work. Additionally, the other reviewers did not raise any ethical concerns.
>
> To clarify, our dataset is created using publicly available video data, and all annotations were performed by human annotators under standard practices without involving sensitive personal information or violating copyright. The dataset does not include location- or nationality-specific attributes that could introduce biases, and the task-specific annotations focus solely on objective event detection.
>
> Given the general nature of your comments, we kindly ask if you could provide more specific details regarding your concerns. This will allow us to address them comprehensively and ensure any potential issues are adequately resolved.

---

> ### Author Response · Authors · 2024-11-24
>
> **Dear Reviewer bJ2T,**
>
> Thank you for your detailed feedback and response. We have uploaded a revised manuscript and appendix addressing all your concerns. Below is a summary of the changes made:
>
> **1. "A review of existing closely related works for the concept of 'fast, frequent, and fine-grained' is not performed..."**
>
> We have added more details of related works on "fast, frequent, and fine-grained" event detection in Section 2 (L140–L157) to better contextualize our contributions and highlight the gaps addressed by our approach.
>
> **2. How do these evaluation metrics contribute to the evaluation of performance?**
>
> A detailed discussion justifying our choice of evaluation metrics and their specific contributions to performance evaluation has been added to Section 5 (L354–L366).
>
> **3. Section 3 remains raw and unclear.**
>
> We have revised Section 3, and we believe the updated version is more refined and comprehensive. Could you kindly let us know if it addresses your previous concerns?
>
> **4. Concerns about:
> (i) Is the dataset limited to tennis, or is it being expanded?
> (ii) Is the benchmark ready to contribute towards a "VideoNet for future LLM’s benchmarking and fine-tuning"?
> (iii) What is the scope of this work?**
>
> **(i)** F3Set is not limited to tennis. It also includes datasets for tennis doubles, badminton, and table tennis, as discussed in Section 3 and detailed in the Appendix. Additionally, we provide a general annotation toolchain that enables domain experts to create F3 datasets for other domains. In this paper, we use tennis as a case study to demonstrate the construction of an F3 dataset and evaluate existing and proposed methods, which can be generalized to other applications.
>
> **(ii)** As noted in L081–L082 of the revised manuscript, F3Set is a potential foundation for advancing multi-modal LLM capabilities in F3 video understanding. However, this direction is a future work.
>
> **(iii)** The scope of this paper includes:
> - Introducing F3Set, a benchmark with datasets featuring over 1,000 precisely timestamped event types with multi-level granularity.
> - Proposing a general annotation toolchain for creating F3 datasets.
> - Presenting F3ED, an end-to-end model for accurate F3 event detection.
> - Conducting comprehensive evaluations and ablation studies of leading temporal action understanding methods on F3Set.
>
> We hope these updates address your concerns. If further clarification or additional revisions are needed, we are willing to provide more information.

---

> > ### Author Response · Authors · 2024-11-26
> > **Follow-Up on Revised Manuscript and Ethical Concerns**
> >
> > Dear Reviewer bJ2T,
> >
> > We would like to follow up on our revised manuscript and the explanation addressing the ethical concerns you raised. Specifically, we hope to confirm whether the revisions and clarifications adequately address your concerns. If there are any remaining issues or additional feedback, we would be happy to address them promptly.
> >
> > Regarding your third concern, "Section 3 remains raw and unclear," we have significantly revised this section and believe it is more refined and comprehensive. Could you kindly let us know if the updated version resolves your previous concerns?
> >
> > Thank you once again for your time and suggestions.

---

> ### Author Response · Authors · 2024-11-27
> **Follow-Up on Revised Manuscript and Ethical Concerns**
>
> Dear Reviewer bJ2T,
>
> We wanted to kindly follow up to inquire whether our response and the revised manuscript have adequately addressed your concerns. As the deadline for **uploading the final revised PDF approaches soon** (27 November), we would greatly appreciate knowing if you have any further suggestions or feedback on our revisions. Thank you again for your time and thoughtful review. We look forward to hearing from you.
>
> Best regards,
> The Authors

---

> ### Author Response · Authors · 2024-11-29
> **A gentle reminder for discussion**
>
> Dear Reviewer bJ2T,
>
> We would like to kindly follow up on whether our revised manuscript and responses have adequately addressed your concerns. Below, we **recap** your previous concerns and how we have addressed them:
>
> **1. A review of existing closely related works for the concept of "fast, frequent, and fine-grained."**
>
> We have added more details of related works on "fast, frequent, and fine-grained" event detection in Section 2 (L140–L157), contextualizing our contributions and highlighting the gaps our approach addresses.
>
> **2. How do these evaluation metrics contribute to the evaluation of performance?**
>
> A detailed justification of our evaluation metrics and their specific contributions to performance evaluation has been included in Section 5 (L354–L366).
>
> **3. Section 3 remains raw and unclear.**
>
> We have revised Section 3, and we believe the updated version is more refined and comprehensive. Could you kindly let us know if it addresses your previous concerns?
>
> **4. Concerns about:**
>
> (i) **Dataset Scope:** F3Set includes datasets for tennis singles, doubles, badminton, and table tennis, as discussed in Section 3 and detailed in the Appendix. A general annotation toolchain enables the creation of F3 datasets for other domains, with tennis serving as a case study in this paper.
>
> (ii) **VideoNet Contribution:** As noted in L081–L082, F3Set is a potential foundation for advancing multi-modal LLM capabilities in F3 video understanding. However, this direction is a future work.
>
> (iii) **Scope of Work:** The scope of this paper includes:
> - Introducing F3Set, a benchmark with datasets featuring over 1,000 precisely timestamped event types with multi-level granularity.
> - Proposing a general annotation toolchain for creating F3 datasets.
> - Presenting F3ED, an end-to-end model for accurate F3 event detection.
> - Conducting comprehensive evaluations and ablation studies of leading temporal action understanding methods on F3Set.
>
> **5. Ethical Concerns:**
> - The dataset is licensed under CC BY 4.0 International ([link](https://creativecommons.org/licenses/by/4.0/legalcode)) and does not require human ethics approval. It is based on publicly available videos featuring professional players in broadcast tournaments. Similar datasets [1–6], focusing on publicly available sports broadcasts with visible human faces, exist and do not employ anonymization techniques, as these players are already in the public domain.
>
> - The dataset is sourced from publicly available YouTube videos, with links provided in our repository (e.g., [here](https://github.com/F3Set/F3Set/blob/main/data/f3set-tennis/videos.csv)). In cases where a video becomes unavailable (e.g., due to deletion or policy changes), we maintain local copies and can provide them upon request.
>
> - Regarding the dataset maintenance plan, we ensure ongoing availability and usability of the dataset by archiving all video files locally and providing direct access to annotations through our GitHub repository ([link](https://github.com/F3Set/F3Set/tree/main/data)). Should videos become unavailable online, our local archives serve as a backup for researchers. Additionally, we plan to periodically review and update the dataset to include new data from emerging tournaments, ensuring its relevance and comprehensiveness. We also encourage community contributions to expand the dataset’s scope under the same licensing terms.
>
> - We plan to release the full dataset, including tennis singles, doubles, badminton, and table tennis. Annotations for all datasets are already available in our GitHub repository. We also note that Reviewer YGio thought there were no ethical issues with our proposed dataset, as it is licensed under CC BY 4.0 International and based on publicly available content.
>
> ---
>
> We hope these revisions and clarifications adequately address your concerns. If you feel our revisions have satisfactorily resolved your feedback, we would greatly appreciate it if you could consider lifting the score.
>
> Thank you again for your time and feedback. We look forward to hearing from you.
>
> [1] "FineGym: A Hierarchical Video Dataset for Fine-grained Action Understanding." CVPR 2020.
> [2] "SoccerNet-v2: A Dataset and Benchmarks for Holistic Understanding of Broadcast Soccer Videos." CVPR 2021.
> [3] "FineDiving: A Fine-grained Dataset for Procedure-aware Action Quality Assessment." CVPR 2022.
> [4] "Sports video analysis on large-scale data," ECCV 2022.
> [5] "Spotting Temporally Precise, Fine-Grained Events in Video." ECCV 2022.
> [6] "ShuttleSet: A Human-Annotated Stroke-Level Singles Dataset for Badminton Tactical Analysis." KDD 2023.

---

> > ### Comment · Reviewer_bJ2T · 2024-11-29
> >
> > I acknowledge the authors' efforts in providing a detailed revision.
> >
> > After carefully considering the strengths of the paper, the revisions made, and the authors' efforts, the reviewer is pleased to raise the score from 3 to 5.
> >
> > The authors should take all reviewers' suggestions into account in the final revision, while also ensuring that any potential ethics or copyright issues are addressed, if applicable.

---

> ### Comment · Reviewer_bJ2T · 2024-11-29
>
> Regarding "The dataset is sourced from publicly available YouTube videos, with links provided in our repository (e.g., ...). In cases where a video becomes unavailable (e.g., due to deletion or policy changes), we maintain local copies and can provide them upon request." and "Should videos become unavailable online, our local archives serve as a backup for researchers."
>
> While people freely upload material to YouTube, the rights issues can be complex.
>
> If the rights to derivatives are reserved, permission must be obtained from the rights holder.
>
> Therefore, if a video (e.g., one deleted by the original provider) is stored locally and redistributed, it could lead to legal issues.
>
> Some of the concerns I would encourage authors to check:
>
> 1. **Privacy and consent**: The dataset includes publicly available videos featuring professional players. However, even though they are public figures, their participation in the dataset might not have been explicitly consented to for research or redistribution purposes. Using publicly available content without explicit consent from individuals may violate ethical standards, particularly if the data is used for commercial purposes or beyond its original intent.
>
> 2. **Data archiving and copyright**: The dataset maintains local copies of videos in case they become unavailable (e.g., due to deletion or policy changes). While this ensures continuity, archiving videos without proper permissions might violate platform terms of service (e.g., YouTube's terms). Storing and redistributing locally archived copies may inadvertently breach copyright or licensing agreements, **especially if the videos were not intended for long-term distribution or offline access**.
>
> 3. **Anonymization and privacy of individuals** The dataset does not use anonymization techniques for visible human faces, assuming players are in the public domain due to their professional status. Failing to anonymize data, especially for professional players, might raise privacy concerns, particularly if sensitive personal information can be inferred from their actions in the videos.

---

> > ### Author Response · Authors · 2024-11-29
> >
> > Dear Reviewer bJ2T,
> >
> > Thank you for raising the score and for highlighting these important concerns. We recognize the complexities surrounding copyright, privacy, and consent for publicly available YouTube videos and have carefully addressed these issues, referencing how similar research datasets manage them (e.g., [Sports-1M](https://cs.stanford.edu/people/karpathy/deepvideo/)).
> >
> > 1. **Privacy and Consent**:
> > Our dataset is based on publicly available videos featuring professional players in widely broadcast tournaments. While these players are public figures, we understand that their participation in the dataset may not have been explicitly consented to for research purposes. However, the dataset is intended solely for academic research and not for commercial use, aligning with the original purpose of these broadcasts—sports analysis and study. We will include a clear disclaimer about its academic use and ensure no sensitive personal information is inferred beyond what is publicly visible in the broadcasts.
> >
> > 2. **Data Archiving and Copyright**:
> > The dataset is distributed as YouTube URLs, ensuring that the original content remains under the control of its rights holders. If a video becomes unavailable (e.g., due to deletion or policy changes), we aim to adhere strictly to the licensing terms of the videos. For example, if a video is shared under a Creative Commons license that allows redistribution, we may provide a copy, ensuring compliance with all legal requirements. However, for videos without such permissions, we will not redistribute local copies and will adjust the dataset annotations accordingly.
> >
> > 3. **Anonymization and Privacy of Individuals**:
> > While the dataset does not anonymize visible human faces, as these players are public figures, we ensure no personally sensitive information is inferred or exploited beyond what is presented in the original broadcast. To further uphold ethical standards, we will explicitly state in the dataset documentation that it is intended for academic research and not for uses that may infringe upon privacy or consent.
> >
> > We will incorporate these considerations into our final revision, carefully addressing all potential ethical and copyright issues raised by reviewers. Thank you once again for your feedback and suggestions.

---

> ### Author Response · Authors · 2024-12-02
>
> Dear Reviewer bJ2T,
>
> As the discussion period nears its conclusion, we wanted to kindly inquire whether our responses have sufficiently addressed your recent concerns regarding privacy, consent, copyright, and dataset handling.
>
> If there are any remaining questions or suggestions, we would be happy to provide further clarifications. Additionally, we would like to respectfully ask whether our explanations and updates might merit a further increase in the score.
>
> Thank you once again for your time and thoughtful review.

---

> > ### Comment · Reviewer_bJ2T · 2024-12-02
> >
> > Dear authors,
> >
> > Thank you for your clarifications and efforts to address the concerns. However, given the potential implications surrounding privacy, consent, copyright, and dataset handling, I believe it is best to defer these matters to the Area Chair, Senior Area Chair, and Program Committee for the final decision.
> >
> > Please note that my increase in score from 3 to 5 is based on the efforts made in the revisions. As such, I will be maintaining my current rating of marginal reject.
> >
> > Best regards,
> > Reviewer bJ2T

---

### Official Review · Reviewer_JNET · 2024-11-04

**Soundness:** 3
**Presentation:** 3
**Contribution:** 3
**Rating:** 8
**Confidence:** 4

**Summary:**

Thanks the authors for this ICLR2025 submission.

Summay:

This paper makes contribution to faciliating the detection of fast-speed, high frequence and fine-grained action events in videos. A new dataset, specific to **tennis**, is proposed to this purpose, together with an video frame-wise label annotation tool, which enables the collections of accurate labels. In experiments, a novel model is proposed and benchmarked on temporal action detection over 3 levels of granularity. Some ablation study is provided. To showcase the scalabiility and generalizability of proposed apporach, the paper extends the study to several other action-sports datasets (e.g., diving, gym, etc).

**Strengths:**

- Writing is easy to follow.
- The proposed dataset is collected with notable efforts, and is very-well organzied and presented. So does the 3-step annotation procedure, a very practical approach to the video dense annotation problem.
- The appendix further provides comprehensive stats.
- The experimental results are strong.

**Weaknesses:**

In general, there are a few minor weakness spotted in the experiment section which puts question marks to the technical sound of this paper.

- (minor) The statement on line 398-399, `...it is crucial to utilize frame-wise feature extraction [7]`, is not well supported. The author might have gained much insights through some hidden experiments that clip-wise feature extration is inferior to frame-wise methods. Yet, it is less clear to the general audience.
- (minor) It would be more insightful to provide the numeric impact of the **Event localizer** to the whole F^3ED system. As it would be a concern if a very good performing LCL module is the **hard-prerequisite**, in which case the generalization of the F^3ED approach is bit questionable. In some scenarios,  a good LCL might not be possible to have.
- (minor) There is a lack of in-depth discussion why GRU based method is adopted in the CTX stage, given the transformer module has been demonstrated more efficient on long-range context modeling.
- (minor) Figure 4 should be more clear on 1) what is the "plus" symbol that combines outputs of LCL and MLC, and 2) what are the red squres in feature vectores under the CTX module.

Missing reference:
- Sports video analysis on large-scale data, ECCV 2022

**Questions:**

see weakness.

---

> ### Author Response · Authors · 2024-11-20
> **Rebuttal by Authors**
>
> Dear reviewer,
>
> We thank your valuable feedback and comments and have carefully considered them to make the following responses:
>
> **Q1: The statement on line 398-399, ...it is crucial to utilize frame-wise feature extraction [7], is not well supported...**
>
> A1: Thank you for raising this concern. We appreciate the chance to clarify the advantages of frame-wise over clip-wise feature extraction.
> - Temporal Precision: To use a clip-wise feature extractor, we can divide the input video into non-overlapping segments and extract one feature vector per segment, which is a common technique in many TAL and TAS tasks. To investigate this, we conducted an experiment where 96-frame video clips were divided into 6-frame segments, with features extracted using I3D [1] for each segment. The resulting 16 feature vectors were interpolated back to 96 frames using PyTorch’s F.interpolate function. As shown in the table below, this approach produces temporally coarse features, leading to inadequate performance in precise event detection tasks.
>
> - Efficiency and Scalability: An alternative approach is to stride a clip-wise feature extractor to obtain per-frame feature densely. However, this approach introduces significant computational overhead as each frame is processed multiple times in overlapping windows. This overhead makes end-to-end feature learning or fine-tuning impractical. In contrast, our frame-wise approach processes each frame only once, enabling the training of much longer sequences (hundreds of frames) in an end-to-end manner on a single GPU.
>
> |Video encoder|Head arch.|F3Set (Ghigh)|||
> |-|-|-|-|-|
> |||F1_evt|F1_elm|Edit|
> |TSM (frame-wise)|GRU|31.4|71.4|68.7|
> |I3D (clip-wise)|GRU|22.7|59.7|64.5|
>
> [1] "Quo Vadis, Action Recognition? A New Model and the Kinetics Dataset." CVPR 2017.
>
> **Q2: More insightful to provide the numeric impact of the Event localizer to the whole F3ED system...**
>
> A2: Thank you for your suggestion regarding the impact of the Event Localizer (LCL) on the performance of the overall F3ED system. We agree that analyzing this relationship is critical for understanding the system's generalization. Therefore, we conducted additional analysis and included an "F1_lcl" column in the table below, which evaluates the precision of the LCL module in identifying event moments with tight temporal tolerance. The table compares the F1_lcl metric with the overall system metrics (F1_evt and Edit) across various datasets:
>
> |Dataset|F1_lcl|F1_evt|Edit|
> |-|-|-|-|
> |F3Set (G_high)|86.7|40.3|74.0|
> |ShuttleSet|97.9|70.7|77.1|
> |FineDiving|94.2|77.6|95.1|
> |FineGym|84.0|70.9|70.7|
> |CCTV-Pipe|71.9|37.0|39.5|
>
> We observe that the performance of F3ED is positively correlated with the quality of the LCL module. For example, datasets like FineDiving and ShuttleSet, which have high-performing LCL modules, result in superior downstream performance (F1_evt and Edit). Conversely, on datasets like CCTV-Pipe, where the LCL module performs less effectively, F3ED’s overall performance is less ideal.
>
> However, it is important to highlight that even when the LCL module does not perform well, our method still outperforms other state-of-the-art methods (as shown in Table 4 in the paper). Therefore, a very good-performing LCL module is not a hard prerequisite. We will incorporate this analysis into the paper.
>
> **Q3: Lack of in-depth discussion why GRU-based method is adopted in the CTX stage, given the transformer module has been demonstrated more efficient on long-range context modeling.**
>
> A3: Thank you for your constructive feedback. We acknowledge that transformer-based models have demonstrated superior efficiency in modeling long-range dependencies. To ensure our choice was justified, we conducted comparative experiments using a Bidirectional GRU (BiGRU) and a Transformer Encoder for the CTX stage. The results are summarized in the table below:
>
> ||F3Set (Ghigh)|||F3Set (Gmid)|||F3Set (Glow)|||
> |-|-|-|-|-|-|-|-|-|-|
> ||F1_evt|F1_elm|Edit|F1_evt|F1_elm|Edit|F1_evt|F1_elm|Edit|
> |F3ED (CTX: GRU)|40.3|75.2|74.0|48.0|76.5|82.4|68.4|80.0|87.2|
> |F3ED (CTX: Transformer)|39.0|74.3|72.8|50.5|75.5|81.8|63.4|79.6|86.8|
>
> As the results indicate, the performance of the two module choices is comparable, with BiGRU slightly outperforming the Transformer Encoder in our F3ED system. We attribute this to the relatively short event sequences passed to the CTX module, which typically contains fewer than 20 events per 96-frame input clip. Under these conditions, the BiGRU effectively models the necessary temporal context with fewer parameters and lower computational overhead compared to the Transformer Encoder. We will incorporate this discussion into the paper to provide additional clarity on the rationale behind our choice.

---

> ### Author Response · Authors · 2024-11-20
> **Rebuttal by Authors**
>
> **Q4: Figure 4 should be more clear on 1) what is the "plus" symbol that combines outputs of LCL and MLC, and 2) what are the red squares in feature vectors under the CTX module.**
>
> A4: Thank you for your valuable suggestions regarding Figure 4. We apologize for the lack of clarity and appreciate the opportunity to improve the figure's description.
> - Clarification of the "plus" symbol: The "plus" symbol represents the combination of outputs from the Event Localizer (LCL) and the Multi-Label Classifier (MLC). Specifically, this operation merges the results of the LCL, which identifies event timestamps, with the outputs of the MLC, which generates multi-label representations for each frame. The combined output is a sequence of multi-label event representations at identified event moments, which is then passed to the CTX module for further refinement.
> - Explanation of the "red squares": The red squares in the feature vectors under the CTX module highlight incorrectly predicted elements in the multi-label event representations. These errors may occur when the initial event detection relies solely on visual features, which can sometimes lead to inaccurate predictions.
>
> We will include a more detailed caption in Figure 4 for greater clarity.
>
> **Q5: Missing reference.**
>
> A5: Thank you for pointing out the missing reference on team sports video analytics. We appreciate the reminder to include this important work. We will add the missing reference, along with other similar works in Section 2 by including a sentence: *"Several datasets also focus on team sports video analysis. For example, Volleyball [1] and NSVA (basketball) [2] emphasize group activity understanding and video captioning, while SoccerNet [3] encompasses a broader range of tasks, including action spotting, camera calibration, player re-identification, and tracking in soccer."*
>
> [2] "A hierarchical deep temporal model for group activity recognition." (CVPR 2016).
> [3] "Sports video analysis on large-scale data." (ECCV 2022).
> [4] "Soccernet-v2: A dataset and benchmarks for holistic understanding of broadcast soccer videos." (CVPR 2021).

---

> ### Author Response · Authors · 2024-11-22
> **A gentle reminder for discussion**
>
> Dear Reviewer JNET,
>
> We sincerely appreciate the time you have dedicated to reviewing our paper and providing valuable feedback. We have carefully addressed your concerns and uploaded a revised manuscript with all changes highlighted in blue. To accommodate the page limit, some of your concerns have been addressed in the Appendix, which is also highlighted in blue.
>
> We would be grateful if you could kindly review our rebuttal. If any points require further clarification or elaboration, we would be happy to provide additional details. Your prompt response would be greatly appreciated.
>
> Thank you once again for your time and thoughtful consideration.
>
> Best regards,
> The Authors

---

> ### Author Response · Authors · 2024-11-24
> **A gentle reminder for discussion**
>
> Dear Reviewer JNET,
>
> Thank you for your detailed comments on our work. Your feedback helped us improve our paper's quality. We would be happy to discuss any further questions and comments you may have. Please let us know if you have received our responses and if we have successfully addressed your concerns.
>
> Best regards,
>
> The Authors

---

> ### Comment · Reviewer_JNET · 2024-11-25
> **Thanks**
>
> Thanks the authors for the feedback, which mostly addressed my questions.
>
> The question that lingers with me now is what makes this dataset so distinctive, such that more advanced techniques in other fields, such as I3D / SlowFast / Transformers, perform worse than simpler ones here.  Were the experiments done with the same configurations, e.g., batch-size, number of parameters, etc?
>
> Also, I'd like to suggest holding the scope of discussion in lines 425-429 in the updated manuscript to F^3Set dataset: For example consider change `..., indicating that increasing the video encoder’s complexity does not necessarily translate to better performance` to  `..., indicating that the complexity of video encoder might not scale well to the fast action detection performance of F^3Set dataset`.
>
> Another suggestion is to present the paper in the following way: Showcase the failure of stonger models (e.g., I3D / Transformers) -> analyze and demonstrate what makes them ineffective (e.g., difficulty in optimization in form of plateaued loss curves, unable to capture important visual details in single frames, or generally worse performance over longer videos) -> propose simpler but more effective solution (e.g, F^3ED) that can address their weakness.
>
> Having seen all the other discussions, I would like to hold on to my original score.

---

> > ### Author Response · Authors · 2024-11-25
> >
> > Dear Reviewer JNET,
> >
> > Thank you for your thoughtful follow-up and suggestions, which provide valuable directions for improving our work. We address your specific concerns below:
> >
> > **Q1: What makes this dataset so distinctive, such that more advanced techniques perform worse than simpler ones here?**
> >
> > Thank you for raising this important question. We ensured that all experiments were conducted under consistent configurations, including batch size (4), input image sequence length (96 frames per clip), stride size (2), resolution (224×224), training epochs (50), and data augmentation techniques. Detailed model configurations are provided [here](https://drive.google.com/drive/folders/1SnHkYFWvt4v9E0m5j3J1GbjNH6b0NNuk). Notably, for the transformer-based method (VTN), the batch size had to be reduced to 2 due to its high computational requirements on the GPU. For transparency, the table below lists the number of parameters and learning rates for each model, adjusted to maximize performance based on their complexities.
> >
> > | **Model**          | **Parameters (M)** | **Learning Rate** |
> > |---------------------|--------------------|-------------------|
> > | I3D                | 22.3               | 0.001             |
> > | SlowFast           | 49.4               | 0.0001            |
> > | VTN                | 102.7              | 0.0001            |
> > | 2D CNN (RegNet-Y) + TSM| 5.6                | 0.001             |
> >
> > **Performance Differences:**
> > Two key factors explain why advanced models perform worse on F^3Set:
> >
> > 1. **Nature of F^3Set Events**:
> >    Events in F^3Set are extremely fast and short, typically spanning only 1–2 frames. Effective detection of such events requires video encoders that capture fine-grained spatio-temporal features between adjacent frames. Advanced models like I3D, SlowFast, and transformers are designed for longer-duration actions (spanning several seconds) and often using non-overlapping snippets or downsampled frames (altough SlowFast has a densely sampled fast pathway, it uses a much simpler 3D CNN for dense feature extraction), which produce temporally coarse features. This makes them less suitable for detecting rapid, short-duration events in F^3Set.
> >
> > 2. **Strength of the 2D CNN + TSM Approach**:
> >    While simpler, our 2D CNN + TSM method incorporates a state-of-the-art RegNet-Y backbone, a compact and advanced 2D CNN architecture that effectively captures detailed visual information. We used RegNet-Y 200MF by default, but increasing its complexity to RegNet-Y 800MF further improved performance. This demonstrates that success in F^3Set requires a combination of dense feature extraction and sufficient expressiveness to capture subtle visual cues.
> >
> > **Q2: Suggestion to revise lines 425–429 to focus on F^3Set’s unique characteristics.**
> >
> > Thank you for this suggestion. We agree that emphasizing this observation as unique to the F^3Set dataset will improve clarity. Accordingly, we have revised the manuscript to reflect this point.
> >
> > **Q3: Suggestion to structure the paper by showcasing the failure of stronger models, analyzing their weaknesses, and proposing a simpler but more effective solution.**
> >
> > We appreciate this insightful suggestion. This revised structure enhances readability and strengthens the narrative. In the updated manuscript, we have restructured the experimental section to first highlight the limitations of stronger models, analyze their inefficacy in F^3Set, and then showcase the strengths of our proposed method. These changes are reflected in L411–L424.
> >
> >
> > We hope our responses address your concerns. While we understand your decision to maintain your original score, we hope the additional clarifications and manuscript updates might prompt you to reconsider. Please let us know if there are further points to address.

---

> ### Comment · Reviewer_JNET · 2024-11-25
>
> Thanks again for the prompt response.
>
> Given the small batch sizes, it seems that the computational cost might be a major blocker for your research (correct me if I am wrong). In current date of research, there are definitely multiple essential factors affecting the downstream task performance and the associated research conclusions - Optimization, model design, data quality, etc. To marginalize the impact of each, it requires **A LOT OF** computes (e.g., sweeping through hyper-parameters to rule out the optimization, and model architectures) . And the reviewer grab a sense that your research facilities might not be strong enough to thoroughly study the end goal very well. For example, I can imagine some other team can outperform the F^3ED results by modifying the frame sampling strategy or rate of I3D / SlowFast / TimesFormer for better results. Once again, please consider constrain the scope of discussion throughout the paper to 1) the proposed dataset, 2) task and 3) computational resources (e.g., highlight all experiments are done on XXX GPUs).
>
> Besides, the reviewer would like to encourage authors to show the full picture of experiments (e.g., lack of computes for larger batch sized and frames, comparison of parameters, loss curve behaviours, etc.) to the general audience. The truthfulness of experiments details will deliver a very good knowledge to the entire research community in term of what has been done, what is missing and what might be the causes. And it probably will also make it clearer to you what's the next best future research action. This work is more about experimental endeavors and not so much about theory. So, showing **what it is and what it is not** is very helpful and critical.
>
> Considering the efforts on collecting the dataset and trying the best given limited computational resources, I would like to lift my score by 1. Please make sure to include above key points in revision.
>
> In terms of the dataset ethical issues, the reviewer is not an expert, and have to leave it for other reviewers and ACs. If the dataset violates many important rules, I am ok with reconsider my score.

---

> > ### Author Response · Authors · 2024-11-26
> >
> > Dear Reviewer JNET,
> >
> > Thank you for raising your score and for providing thoughtful suggestions. We appreciate your acknowledgment of the challenges involved in conducting extensive experiments to explore factors such as optimization, model design, and data quality on downstream task performance. Below, we address your comments in detail:
> >
> > **General comment on computational constraints and research scope.**
> >
> > We recognize that thorough exploration of factors like optimization, model architecture, and hyper-parameters often requires extensive computational resources, including multiple GPUs. However, our research focuses on the unique F^3 task—a critical yet underexplored challenge in real-world applications. To ensure accessibility for the broader research community, all experiments were conducted on a single GPU.
> >
> > Our experimental setup evaluates widely used configurations for popular video encoders, such as SlowFast, I3D, and TSM, to identify the most critical factors affecting F^3 event detection. For F^3ED, we carefully balanced accuracy and computational efficiency by selecting RegNet-Y with TSM as our video encoder. This serves as a baseline for future development while demonstrating an effective trade-off between performance and computational requirements. Importantly, the key components of F^3ED—the multi-label classifier and contextual module—are **generalizable** and can enhance more complex video encoders. These components improve training efficiency and refine event sequences using contextual knowledge, mitigating errors like logical inconsistencies or rare cases that visual cues alone may fail to resolve.
> >
> > In response to your feedback, we have revised the manuscript to ensure the scope of discussion remains focused on the F^3 dataset, task, and computational considerations.
> >
> > **Full picture of experiments (e.g., computational limitations, parameter comparisons, loss curves)**
> >
> > Thank you for emphasizing the importance of providing a comprehensive view of our experiments. We agree that transparency is vital for advancing community understanding. Due to main text page constraints, detailed experimental setups—including batch sizes, frame counts, training parameters, and the GPU configuration—are presented in **Appendix E** "Implementation Details". We have added more details in the updated manuscript and are prepared to include additional information, such as loss curve behaviors and parameter comparisons, to provide a deeper understanding of what has been achieved, what remains unexplored, and the potential causes of performance differences.
> >
> > We appreciate your recognition of our efforts. By clarifying the scope, computational setup, and results, we aim to provide meaningful and accessible contributions to the broader research community. Thank you again for your thoughtful feedback and for raising the score.

---

### Meta-Review · Area_Chair_sxWD · 2024-12-16

**Metareview:**

This paper makes contribution to faciliating the detection of fast-speed, high frequence and fine-grained action events in videos. Writing is easy to follow. The proposed dataset is collected with notable efforts, and is well-organzied. So does the 3-step annotation procedure, a very practical approach to the video dense annotation problem. The experimental results are strong. The reviewers also recognized that the experimental part of this paper was not clearly presented, the experimental results lacked in-depth discussion, the dataset contribution was limited, and the writing was sloppy. After rebuttal, the reviewer bJ2T improved the score. So the final vote is Accept.

**Additional Comments On Reviewer Discussion:**

The author is asked to carefully improve the writing and clarification of the paper as requested by the reviewer.

---

### Decision · Program_Chairs · 2025-01-22

Accept (Poster)